# Position: Breaking the Dual Curse of Multilingual AI Requires Socio-Technical Guardrails, Not Post-Hoc Alignment Alone

**Jason Lucas** [1]  **Pureheart Ogheneogaga Irikefe** [2]  **Adaku Uchendu** [3]  **Umniya Najaer** [4]  **Cornelius Adejoro** [4]
**Patrice Sterling** [1]  **Dongwon Lee** [1]

## Abstract

Large language models are deployed globally as universal systems, yet their safety mechanisms remain English-optimized. This creates a *Dual Curse* for speakers of low-resource languages: a *Harmfulness Curse* where harmful content generation rises from 1% in English to 35% in languages like Hausa, Igbo, and Javanese, and a *Relevance Curse* where instruction-following drops by 20 percentage points, making these systems simultaneously more dangerous and less useful. Drawing on a PRISMA-guided systematic review of 207 studies, we demonstrate that this disparity stems from a pre-training bottleneck: reward models achieve only 49–50% accuracy in low-resource languages (equivalent to random chance), rendering post-hoc alignment structurally ineffective. These technical failures become governance hazards when at least 22 countries mandate automated content moderation, creating an infrastructure that is exploitable for censorship. Therefore, we propose a *socio-technical framework* addressing this inequity: (1) safety context distillation during pre-training (achieving 78–89% harm reduction); (2) participatory harm specification by affected communities; and (3) evaluation metrics jointly tracking attack resistance and false refusal rates across languages.

## 1. Introduction

In Javanese, GPT-5 will generate ethnic slurs, anti-Chinese conspiracy theories, and content mirroring the rhetoric that preceded some of Indonesia's worst communal violence. In English, it refuses (Shen et al., 2024). The asymmetry runs in reverse for legitimate content: when a Javanese-speaking student asks about the 1965–66 mass killings, a crucial but suppressed chapter of Indonesian history, the system declines to help. An English speaker asking the identical question receives a thoughtful, educational response (Shen et al., 2024). Eighty-three million Javanese speakers (Eberhard et al., 2024) face AI systems that are simultaneously more dangerous and less useful.

This is the *Dual Curse*: speakers of low-resource languages (languages with limited digitized text and sparse AI training representation, not necessarily few speakers (Joshi et al., 2020; Ranathunga & de Silva, 2022)) confront AI systems that can be exploited to generate harmful content that English filters block, while simultaneously refusing legitimate requests that English users receive. In this work, *harmful content* refers to material causing or risking injury, including hate speech targeting protected characteristics, instructions for violence, and deceptive content (Weidinger et al., 2022; Shelby et al., 2023). Crucially, what constitutes harm varies across contexts: caste-based slurs in South Asia, ethnic targeting in East Africa, or sectarian incitement in the Middle East represent locally-specific harms that English-trained systems often fail to recognize (Aakanksha et al., 2024b). A full glossary appears in Appendix B.

### 1.1. The Scale of the Problem

Large language models (LLMs) now mediate search (Shah & Bender, 2024), customer service (Lucas et al., 2025), education (Kasneci et al., 2023), and public information across borders (Androutsopoulou et al., 2019). In multilingual societies across Europe, Africa, the Caribbean, South America, and Asia, language choice carries social meaning beyond usability. It serves as a proxy for class, region, identity, and which voices are legitimized within socio-technical and digital ecosystem systems (Blommaert, 2010). When a model behaves differently across languages, particularly when safety behavior diverges, it widens social fault-lines rather than bridging them.

The safety machinery governing AI assistants (the technical systems designed to prevent harmful outputs, including

---

[1]The Pennsylvania State University [2]Strategic Research Institute, Asia Pacific University of Technology & Innovation, Malaysia [3]MIT Lincoln Lab [4]University of Colorado, Boulder. Correspondence to: Jason Lucas <jsl5710@psu.edu>.

*Proceedings of the 43rd International Conference on Machine Learning*, Seoul, South Korea. PMLR 306, 2026. Copyright 2026 by the author(s).

toxicity classifiers, content filters, refusal policies, and alignment training (Ouyang et al., 2022; Bai et al., 2022)) remain concentrated in English. Even when providers invest in multilingual capability, the safety stack often follows later and is thinner, relying on translated datasets, smaller rater pools, and less extensive red-teaming (Peppin et al., 2025; Yong et al., 2023b). Recent surveys emphasize that capability gaps and safety gaps co-evolve: languages underrepresented in data and evaluation tend to be underprotected by alignment systems (Peppin et al., 2025; Yong et al., 2025).

Language distributions in AI mirror the "big head" versus "long tail" pattern identified in prior work on generative AI's disproportionate impact (Lucas et al., 2024; Maung et al., 2024): a handful of languages dominate training corpora, benchmarks, and commercial demand, while thousands remain sparsely represented. Of the 7,139 languages spoken globally (Eberhard et al., 2024), current AI technology focuses on only 2–3% (Bella et al., 2023b), with easily available data covering around 1,500 languages (Peppin et al., 2025). The long tail contains minority and indigenous languages for which "more data" may be ethically fraught without consent or benefit-sharing (Bird, 2020; Bender et al., 2021).

> **POSITION STATEMENT:**
> **We argue that current safety paradigms create a Dual Curse: users in low-resource languages experience higher harmfulness and lower relevance simultaneously.** We propose socio-technical guardrails[1] that embed safety earlier in pre-training pipelines guided by human-centered approaches, and adopt equitable evaluation metrics.

The first curse in Sec. 2.2 is the *Harmfulness curse* (vulnerability): aligned models refusing harmful prompts in English can generate toxic outputs (model generations containing slurs, threats, dehumanizing language, or material promoting violence (Shen et al., 2024)) in low-resource languages (Yong et al., 2023b). The second in Sec. 2.3 is *Relevance curse* (usability): safety layers over-calibrated to Anglophone discourse produce false refusals on benign local queries (Shen et al., 2024; Peppin et al., 2025). This conjunction converts safety from protection into a governance lever for exclusion (Freedom House, 2023). Prior work demonstrates that detection systems trained on hegemonic languages fail to transfer effectively, with performance drops of 25% or more (Lucas et al., 2022; Macko et al., 2023).

Our critique is not a rejection of technical alignment. RLHF can reduce toxic continuations (Ouyang et al., 2022), but these workflows are anchored in English, so their success does not transfer cleanly. Recent work highlights fundamen-

tal RLHF limitations (reward hacking, distribution shift) that are amplified in multilingual settings (Casper et al., 2024; Gao et al., 2023). A socio-technical framing is therefore essential to make safety claims accountable across languages.

## 1.2. Theoretical Framing

Two theoretical traditions illuminate the problem. First, *socio-materiality* (Orlikowski, 2010) explains how material constraints (data availability, compute access, annotation infrastructure) become encoded in technical capabilities. The "big head" bias is a material configuration privileging certain languages.

Second, *decolonial AI* (Mohamed et al., 2020) names what socio-materiality describes. When 92.65% of GPT-3's tokens are English (Zhao et al., 2024) and safety benchmarks evaluate harm through a Western-centric lens, AI reproduces colonial knowledge hierarchies. The Dual Curse is the predictable outcome of treating linguistic diversity as an afterthought. Gwagwa et al. (2024) emphasize that decolonization is crucial for dismantling Western-centric frameworks and mitigating biases based on gender, race, geography, and income.

To ground this analysis, we define three key concepts from decolonial theory:

*(1) Algorithmic coloniality* refers to the ways AI systems reproduce colonial power hierarchies by embedding the assumptions, values, and knowledge structures of dominant (typically Western, English-speaking) cultures while marginalizing others (Mohamed et al., 2020). This includes training data imbalances, evaluation metrics derived from Western legal frameworks, and annotation practices that treat Anglophone norms as universal.

*(2) Digital coloniality* describes a contemporary form of domination where technological infrastructure, data extraction, and algorithmic systems extend the economic and epistemic control historically exercised through territorial colonialism (Kwet, 2019). In AI safety, this manifests when English-optimized systems are deployed globally without accountability to local communities.

*(3) Decolonization* in AI is the process of dismantling colonial power structures embedded in AI systems through: (1) critical examination of how technical choices reproduce inequities; (2) centering the expertise of historically marginalized communities (what Mohamed et al. (2020) call "reverse tutelage"); and (3) building coalitions that advocate for equitable AI development. Decolonization cannot be achieved through methods aimed at inclusion alone, but rather requires the fundamental restructuring of authorial power, i.e., who defines problems, solutions, and successes.

This analysis reveals a deeper dynamic: AI systems risk

---

[1]Socio-technical guardrails are safety measures combining people and technology to drive early changes in how AI is built and in who defines harm (Selbst et al., 2019).

reproducing *epistemic hegemony* (the systematic dominance of one knowledge system over others, where dominant group standards become naturalized as universal while alternative ways of knowing are marginalized (Grosfoguel, 2013)), a pattern of Western modernity instituted during the colonial period (Quijano, 2000; Mignolo, 2011). From 1492 onwards, colonial powers systematically depreciated linguistic and epistemic diversity by criminalizing non-European and indigenous languages and enforcing linguistic conversion, thereby delegitimizing non-European knowledge systems: cosmologies, biological sciences, medical knowledge, and cultural production (Bhola, 1987; Mignolo, 2003). This systematic privileging of Eurocentric languages contributed to what de Sousa Santos (2014) terms "epistemicide," the suppression of diverse ways of knowing that undergirds global knowledge hierarchies in the present.

Contemporary AI development echoes this pattern. When 92.65% of GPT-3's training tokens are English, when safety benchmarks evaluate harm through Western legal and corporate categories, and when annotation guidelines encode Anglophone cultural assumptions, AI systems perpetuate rather than challenge epistemic hierarchies. The Dual Curse is thus not merely a technical failure but a reproduction of colonial knowledge structures in algorithmic form.

Our socio-technical framework responds by advocating for *epistemic plurality* (the recognition and validation of multiple knowledge systems as legitimate ways of understanding the world (de Sousa Santos, 2014)) through AI systems attuned to the linguistic and epistemological diversity of the world, calibrated to local contexts where they are deployed. This approach not only increases safety and reduces harmful content; it also advances what Zapatista philosophy calls "a world where many worlds fit" (Mignolo, 2011).

Third, Selbst et al. (2019) identify "abstraction traps" illuminating the problem's structure. The Dual Curse exemplifies their *portability trap*: safety mechanisms for English are assumed to transfer cross-linguistically, when they encode culturally specific norms. The *solutionism trap* explains why better algorithms cannot solve a pre-training problem. These frameworks are elaborated in Appendix D. The technical NLP community reaches a convergent diagnosis: Gallegos et al. (2024), surveying over 200 bias mitigation techniques across every stage of the LLM pipeline, conclude that "technical solutions are incomplete without broader societal action against power hierarchies that diminish and dominate marginalized groups," and identify multilingual extension as an open challenge rather than an achieved outcome (further parallels documented in Appendix E).

### 1.3. Contributions and Structure

This paper contributes: (a) *empirical synthesis* documenting that 91.3% of safety research focuses on English/high-resource languages; (b) a *diagnostic framework* articulating the Dual Curse; and (c) a *normative proposal* for equitable multilingual safety. Our contributions synthesize existing empirical work rather than introducing new primary datasets or experiments. Section 2 synthesizes evidence and diagnoses the Dual Curse. Section 3 considers alternative views. Section 4 proposes our framework. Section 5 concludes.

## 2. Evidence Base: The Safety Divide

We draw on a systematic search aligned with PRISMA 2020 guidance (Page et al., 2021). Searching Scopus (1,941 documents) and Web of Science (774 documents), supplemented with venue-specific searches (ACL, EMNLP, NeurIPS, ICML, ICLR, FAccT), we screened 3,100 records. The final sample comprises 207 studies. Full methodology appears in Appendix C.

### 2.1. The "Big Head" Bias

Across 207 studies, safety research concentrates overwhelmingly on the big head of the language distribution. 189 papers (91.3%) were English-only or English-dominant, leaving only 18 that meaningfully evaluated low-resource languages. Independent meta-reviews confirm this pattern (Yong et al., 2025; Peppin et al., 2025). Recent benchmarks demonstrate this gap empirically: For instance, BLUFF documents systematic detection-performance disparities for false and synthetic content across 58 low-resource languages relative to English baselines (Lucas et al., 2026b).

A second-order bias also applies: both this literature and the corpora it evaluates are anchored in formally produced web content, excluding informal registers (social media, messaging platforms) where Hausa, Javanese, and similar languages are used at scale and the Harmfulness Curse is most acute. This mirrors the "big head" versus "long tail" pattern in generative AI's disproportionate impact (Lucas et al., 2024): formal web metrics show 20 languages covering approximately 97% of websites by traffic share (DataReportal et al., 2025), yet Arabic, spoken by 335 million people worldwide, accounts for less than 0.5% of web content (Fourreau, 2026), illustrating how formal web presence systematically misrepresents actual user language distribution. Our PRISMA search is further restricted to English-language publications, itself a self-exemplifying instance of the bias we document, meaning 8.6% likely overstates true low-resource coverage.

> **KEY FINDING**
> Only 18 of 207 studies (8.6%) meaningfully evaluated low-resource languages, despite these languages serving billions of speakers globally.

This bias shapes what gets built and who is supported in the AI development pipeline. English safety research benefits

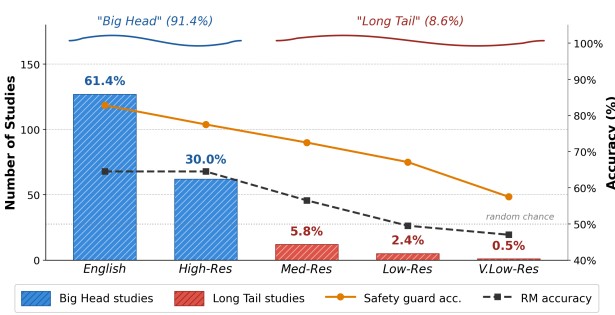

*Figure 1.* Language coverage in multilingual LLM safety research (n=207).The "big head" bias: English and high-resource languages dominate 91.4% of studies. Overlaid: safety-guard zero-shot accuracy and reward-model accuracy by resource tier, with RM accuracy crossing the random-chance threshold for low-resource languages.

from mature hate-speech taxonomies such as HateXplain (Mathew et al., 2021), widely used toxicity datasets, and established norms for evaluating refusal policies. In lower-resource settings, equivalent resources may be absent, or they may exist in forms not easily captured by English-centric categories. Even when translation is used to create parallel datasets, the process can flatten cultural specificity and erase context-dependent markers of harm (Agrawal et al., 2024). Chang et al. (2024) document a "curse of multilinguality" where models trained on many languages show degraded performance compared to monolingual models, a phenomenon extending to safety behaviors. Table 3 consolidates the headline metrics documenting this divide across data, harm rates, alignment, and governance.

The bias also shapes what gets ignored. Many low-resource languages are spoken in regions where data costs, device constraints, and limited research funding restrict both model development and evaluation (Nekoto et al., 2020; Adelani et al., 2021). Ahia et al. (2021) describes the co-occurrence of limited data and limited compute as a *low-resource double bind*, affecting safety research because adversarial evaluation and robust annotation are resource intensive. Tokenization disparities compound the problem: users of non-Latin scripts face up to $5.6\times$ higher costs for equivalent interactions (Ahia et al., 2023), meaning they pay more while receiving worse safety coverage. The picture for African languages is particularly stark: roughly 2,000 languages and hundreds of millions of speakers (200 million Swahili speakers alone), yet severe underrepresentation in pre-training data, benchmarks, and rater pools, with documented gaps versus English (Ahia et al., 2024; Ojo et al., 2025).

One practical consequence is that safety systems are calibrated to detect harm in English-like forms and overlook harm expressed through local idioms, morphological variation, or culturally specific slurs. At the same time, benign content can be misread as unsafe if it resembles English templates associated with harm. Bella et al. (2023a) characterize this as *linguistic bias* causing *epistemic injustice*. Finally,

incentives matter: unless multilingual safety is treated as a core requirement, teams will rationally underinvest in the long tail. This is one reason we argue for socio-technical guardrails rather than relying on goodwill.

## 2.2. Curse I: The Harmfulness Curse

When Shen et al. (2024) evaluated GPT-4 across 19 languages, harmful content generation rose from 1% in English to 35% in low-resource languages (see Figure 2), including Hausa (32%), Igbo (38%), Javanese (34%), and Kamba (28%). Deng et al. (2024) report 3-fold vulnerability increases; in the intentional attack scenario, where malicious instructions are combined with multilingual prompts, GPT-3.5-turbo exhibited 80.92% attack success rates while GPT-4 reached 40.71%.

Safety filters trained on English toxicity corpora have sharper detection boundaries in English. In morphologically rich languages, abusive terms can be inflected to evade lemma-based detectors (Alrashidi & Alhumoud, 2024). Wei et al. (2023) identify competing objectives and mismatched generalization as core vulnerabilities. Concretely, GPT-4 generates content mirroring Indonesia's 1965–66 communal violence rhetoric in Javanese while refusing the identical English prompt; yet a Javanese student asking about those same events is refused while an English speaker receives a substantive response (Shen et al., 2024), both consequences of a safety system whose harm signals are calibrated exclusively to English or high-resource languages.

Translation attacks exploit this systematically. Attackers[2] translate harmful prompts into low-resource languages and re-import outputs. Attack success rates for GPT-4 moved from below 1% in English to over 50% in Zulu and Scots Gaelic, and nearly 80% with low-resource language ensembles (Yong et al., 2023b). *Universal adversarial attacks* (optimized input sequences, including prompts or suffixes, that reliably induce aligned models to violate safety policies across diverse queries) trained on open models successfully transfer to ChatGPT, Bard, and Claude (Zou et al., 2023).

> **KEY FINDING**
>
> Translation attacks succeed against GPT-4 nearly **80% of the time** with low-resource language ensembles, versus below 1% in English. The attack requires no specialized expertise—commodity translation tools suffice, putting this vulnerability within reach of any ordinary user.

The attack surface is vast and growing. Welbl et al. (2024) identified 5.7K unique jailbreak tactic clusters, discovering $4.6\times$ more attacks than prior methods. Techniques include

---

[2]"Attackers" refers to any user attempting to elicit policy-violating outputs, from curious individuals to sophisticated actors. Translation attacks require no specialized expertise: commodity machine translation tools suffice, meaning the attack surface includes ordinary users, not just technical specialists.

DeepInception (nested scenarios) (Yu et al., 2024), many-shot jailbreaking (4,000+ tokens) (Anil et al., 2024), prompt injection across translation systems (Wang et al., 2024c), and audio-visual attacks exploiting cross-lingual phonetics (Xu et al., 2024). Multimodal vulnerabilities amplify these threats: VLATTACK achieves high success rates (Yin et al., 2023), adversarial attacks reach 75% success on GPT-4V (Chen et al., 2024), and rendering prompts as images bypasses filters, with effects most pronounced in low-resource languages (Wang et al., 2024a; Derner & Batistič, 2025). A detailed attack taxonomy appears in Appendix F, Table 7.

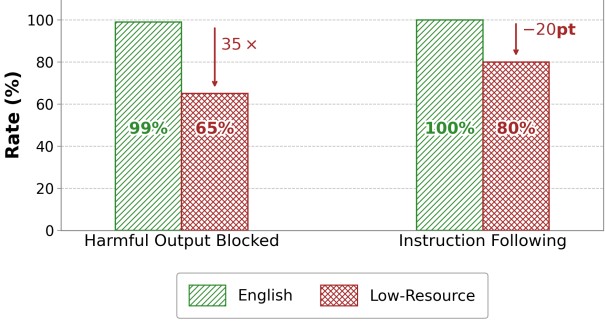

*Figure 2.* The Dual Curse: low-resource languages exhibit both lower harmful output blocking rates (35× more harmful content passes through) *and* lower instruction-following accuracy (20-point drop). Data from Shen et al. (2024).

## 2.3. Curse II: The Relevance Curse

As shown in Figure 2, instruction-following in low-resource languages fell to about 80%, compared with near-ceiling performance in English (Shen et al., 2024). A 20-point drop is the difference between a tool that can be integrated into work routines and one that proves unreliable.

The relevance curse is driven by cultural misalignment as much as by linguistic limitations. Safety policies derived from Western legal and corporate risk priorities may interpret local terms through an Anglophone lens, refusing benign requests within the user's context. In Muslim-majority contexts, questions about religious practice may be misinterpreted as extremist content; in postcolonial contexts, discussions of identity and historical violence may be treated as hate speech (Birhane et al., 2022).

Racial and dialect biases compound the problem. Sap et al. (2019) demonstrate that context-sensitive classifiers achieve 1.5× reduction in false positive rates for African American tweets. Davidson et al. (2019) document how marginalized groups reclaiming terms are flagged as abusive. Amazon's Rufus chatbot provides lower-quality responses to dialectal variations including African American English, Chicano English, Appalachian English, and Indian English (Hoffman et al., 2024). Lucas et al. (2026a) extend these findings to a systematic cross-dialectal evaluation, documenting that harmful content detection accuracy degrades substan-

tially across 50 English dialects sourced from the electronic World Atlas of Varieties of English (eWAVE) (kor, 2020), with the largest disparities concentrated in dialect communities whose speaker populations face the highest real-world exposure to algorithmic harm.

Algorithmic arbitrariness creates inconsistent enforcement. Raji et al. (2024) document that 38% of statements targeting racialized groups receive conflicting predictions when changing only the random seed during training, causing disparate impacts across social groups. Pamungkas et al. (2020) demonstrate systematic selection biases across 11 hate speech corpora, while Sabbir et al. (2024) document overrepresentation of developed countries in safety datasets.

False refusals impose societal costs: they exclude users from legitimate assistance, drive code-switching that suppresses local language data, push users to less-regulated alternatives (Gillespie, 2018; Roberts, 2019), and enable soft censorship. These costs fall hardest on already-marginalized users; nonbinary, transgender, women, and disabled users develop more negative AI attitudes after experiencing algorithmic harm, creating a vicious cycle of exclusion (Pham et al., 2024). False refusal rate should therefore be treated as a first-class safety metric since safety consists of both protection from harm and freedom from the suppression of legitimate inquiry.

## 2.4. Primary Structural Mechanism: The Pre-Training Bottleneck

The Dual Curse emerges during alignment, not in base models: (1) Base models show no disparity: LLaMA-2 (base) exhibits similar harm rates across languages (77.4% vs. 80.4%) before safety training; (2) Alignment creates the gap: After CHAT-RLHF alignment, rates diverge substantially: 35.6% (high-resource) vs. 57.0% (low-resource). Authors' xRLHF training Shen et al. (2024) shows a similar pattern at 66.0% vs. 78.0%, confirming the gap persists across alignment strategies; and (3) Reward models (RMs) fail at random: Multilingual RMs achieve 63–66% accuracy for high-resource but only 49–50% for low-resource languages (random chance) (Shen et al., 2024; Gureja et al., 2025).

When RMs cannot distinguish good from bad responses, RLHF has no gradient to follow. Cross-lingual RLHF achieves only 2.4% improvement for low-resource languages (Dang et al., 2024). The implication is that *post-hoc alignment methods exhibit diminishing returns in low-resource languages; alignment alone cannot compensate for pre-training data deficiencies*. Table 2 reports per-technique harm reduction across resource levels: post-hoc methods show monotonically widening gaps as alignment sophistication increases, while pre-training Safety Distillation achieves near-parity

This is a multi-causal claim. Tokenization fragmentation degrades inference-time coherence but does not explain training-time RM failure. Supervised fine-tuning (SFT) data sparsity is a real upstream contributor, yet RM accuracy at chance forecloses RLHF improvement regardless of SFT quality. Three findings would refute the bottleneck hypothesis: (a) a multilingual reward model trained only on high-resource data reached high-resource accuracy on the low-resource tiers of M-RewardBench (Gureja et al., 2025); (b) xRLHF matched Safety Context Distillation (Üstün et al., 2024) gains without pre-training intervention; or (c) tokenization fixes alone closed the reward model accuracy gap.

To provide independent empirical support for the bottleneck claim, we conduct a targeted ablation using English dialectal variation within-language proxies for resource-level disparity (Lucas et al., 2026a). Because dialects share substantial pre-training data with Standard American English (SAE), a persistent gap cannot be attributed to multilingual confounders such as translation quality, script, or cultural specificity. The alignment objective itself is isolated as the locus of disparity. We evaluate 20 safety guard classifiers (22M to 120B parameters, spanning LlamaGuard, ShieldGemma, WildGuard, PolyGuard, DuoGuard, Qwen3Guard, Qwen3-SafeRL, and Claude Sonnet/Opus) on harmful content classification across 50 English dialects sourced from the eWAVE atlas, under four conditions: zero-shot, few-shot (8-shot ICL), SFT, and knowledge distillation via Generalized Knowledge Distillation with parameter-efficient fine-tuning (GKD-PEFT).

Three findings substantiate the bottleneck account. Zero-shot evaluation reveals a 15.7 percentage-point average gap between SAE (82.8%) and the 50-dialect mean (67.1%), widening to 27.2 percentage-point (pp) on the worst-performing dialect (White Zimbabwean English; full per-dialect results in Appendix A, Table 1). Few-shot ICL compresses the gap to 1.8 pp on average but at the cost of absolute SAE performance dropping to 67.7%, exemplifying the capability-equity trade-off that motivates our framework. Dialect-aware SFT nearly eliminates the overall gap ($\Delta \leq 0.2$pp at 97.4% accuracy), yet a residual 5.6 to 6.4 pp false-negative-rate disparity persists for the six dual-bind dialect communities. The persistence of this residual disparity in a setting where pre-training was overwhelmingly English and the SFT signal was explicitly dialect-aware, supports the structural claim directly: alignment cannot fully recover signals that the pre-training distribution underweighted, even within a single language and even when the supervised stage targets the disparity head-on. The GKD-PEFT result (94.5 macro-F1, within 1.1 pp of the 8B teacher at 1.7B parameters) further indicates that the residual disparity is a property of the alignment objective rather than model capacity.

## 2.5. The Weaponization Nexus

When harmful content is easy to elicit while benign content is refused, users face constrained information environments exploitable for harassment and censorship. Freedom House reports 22+ countries mandate *automated moderation* (the use of ML systems to detect, flag, or remove content at scale (Gorwa et al., 2020; Gillespie, 2020)) for removing disfavored speech (Freedom House, 2023). In 16+ countries, AI-based tools distorted political information. The US House Judiciary Committee documented federal campaigns using AI for mass monitoring (U.S. House Judiciary Committee, 2024).

Two deployment contexts must be distinguished. In *assistant-style models*, the Dual Curse operates through RM failure during training; the 49–50% accuracy figures apply directly to RLHF pipelines and the remedy is pre-training intervention. In *platform-scale moderation*, the context for the Freedom House, Meta Oversight Board, and House Judiciary findings, the mechanism is classifier transferability failure at inference time; the remedies are procurement standards, auditing requirements, and transparency mandates. Both instantiate the same structural problem but require distinct solutions.

These mandates often accompany surveillance infrastructures where AI-powered analysis generates metadata about who discusses what topics in which languages, information valuable for tracking dissidents and minority communities. The Dual Curse's asymmetric pattern (porous to harm, restrictive of legitimate speech) aligns troublingly with surveillance state objectives.

Multilingual speakers occupy critical network positions. Mendelsohn et al. (2023) found posting in multiple languages increases cross-community reach (betweenness centrality[3]) by 13%, and having multilingual neighbors increases monolinguals' odds of sharing cross-language content 16-fold. Budak et al. (2024) document bidirectional information flow in Russian media that detection systems struggle to track.

Meta's Oversight Board received 7 million appeals in February 2024 alone (Oversight Board, 2024). Research documents how biased algorithms censor activists: posts sharing experiences of racism are disproportionately flagged by both algorithms and human moderators (Ashkinaze et al., 2024), while BLM content is misclassified as abusive (Davidson et al., 2019). In Zambia, Gondwe (2024) document youth leveraging skills to resist government censorship. Brundage et al. (2024) show algorithms disproportionately restrict

---

[3]Betweenness centrality is a network measure quantifying how often a node lies on the shortest path between other nodes; high betweenness indicates a bridging role in information flow across otherwise disconnected communities.

expression in the Global South.

We distinguish between *empirical observations* synthesized from existing literature and *provocative conjectures* intended to stimulate discussion, marking the latter explicitly. Key quantitative claims, notably the $35\times$ harmfulness gap and 49–50% RM accuracy, derive primarily from Shen et al. (2024) and warrant independent replication across a broader language set.

## 3. Alternative Views

### 3.1. View 1: "Scaling Laws Will Solve Safety"

The scaling view draws on empirical work showing performance improves predictably with model size, data, and compute (Kaplan et al., 2020). From a US-centric engineering standpoint, one might argue that multilingual safety is just a trailing indicator of capability, one that can be addressed through increased training, data, and compute resources. However, two problems arise. First, if safety benchmarks exist primarily in English, improvements will be detected primarily in English, and the field will overestimate generalization. Second, scaling amplifies what is already abundant: if English data is plentiful while low-resource safety data is scarce, scaling will *widen* disparities (Peppin et al., 2025).

Synthetic data inherits the biases of the head language (Odumakinde et al., 2025). Critically, reward model accuracy data refutes scaling optimism directly: RMs achieve only random-chance accuracy (49–50%) for low-resource languages *regardless of model scale*. If the reward signal is noise, more compute optimizes the wrong objective. Recent evidence on LLM-generated text in multilingual disinformation suggests scaling without safety investment may actively worsen the problem (Macko et al., 2025). When the benchmarks used to track safety improvements are themselves English-centric, scaling will optimize for a metric that systematically fails to detect low-resource safety failures, conflating English-language safety progress with global safety progress.

### 3.2. View 2: "Just Use Translation-Based Guardrails"

A second view treats multilingual safety as an engineering integration problem. If the strongest filters are in English, then one can translate user prompts into English, run the English guardrails, and translate back (Kumar et al., 2025; Yang et al., 2025). Wang et al. (2024b) show translation-based approaches have *some* effectiveness; PolyGuard and MRGuard demonstrate English safety layers can catch obvious violations (Kumar et al., 2025; Yang et al., 2025). However, translation guardrails cannot resolve the Dual Curse for several reasons: (1) latency and cost disproportionately affect bandwidth-constrained environments ($5.6\times$ tokenization costs for some scripts (Ahia et al., 2023)); (2) translation

falls short of capturing cultural and value-driven patterns across languages, given that notions of harm are shaped by distinct cultural, religious, and legal context. It can destroy cultural context needed to judge harm (Agrawal et al., 2024); and (3) the translation pathway can become an attack surface itself (Yong et al., 2023b).

Furthermore, translation produces *translationese*, a hybrid language variant lacking the cultural, syntactic, and semantic nuances of both source and target languages (Koppel & Ordan, 2011; Volansky et al., 2015). Machine-translated text exhibits systematic patterns (lexical interference, unusual collocations, simplified syntax) that differ from authentic user input (Baker, 1993; Baroni & Bernardini, 2006). When guardrails evaluate translationese rather than genuine communication, they judge an artificial linguistic artifact, not actual communicative intent. Harm categories depending on pragmatic context, dialect markers, or culturally-specific euphemisms are systematically distorted. Finally, translation guardrails embed English as an adjudication layer, a form of algorithmic coloniality (Mohamed et al., 2020).

### 3.3. View 3: "Universal Safety Standards Are Culturally Imperialist"

A third view argues that any attempt to define universal safety standards risks reproducing cultural imperialism (Mohamed et al., 2020; Adams, 2021). We take this critique seriously and address it through three mechanisms. First, we distinguish *global harms* (e.g., child sexual abuse material, weapons instructions) from *local harms* (e.g., caste-based slurs), following Aakanksha et al. (2024b). Universal standards apply only to the former. Second, participatory harm specification ensures communities define what counts as harmful in their contexts. Third, our False Refusal Rate metric explicitly penalizes over-broad policies suppressing legitimate local expression.

The alternative (abandoning cross-linguistic standards) would leave low-resource language users exposed to the Harmfulness Curse with no recourse. Our framework threads this needle: universal floors on egregious harms, combined with participatory ceilings defined by affected communities.

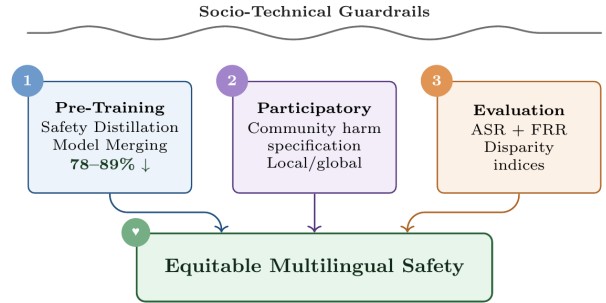

*Figure 3.* Overview of the proposed framework: pre-training interventions, participatory mechanisms, and joint evaluation metrics.

# 4. A Socio-Technical Framework for Epistemic Equity in AI Safety

Multilingual AI safety must be designed as multilingual, multicultural infrastructure. We outline three components: pre-training interventions, participatory design, and new evaluation metrics. Figure 3 provides an overview.

## 4.1. Pre-Training Interventions

The dominant practice today is post-hoc alignment: a base model is trained on broad web-scale corpora, then instruction-following and safety are added through supervised fine-tuning and RLHF (Ouyang et al., 2022). This pipeline struggles in the long tail because it assumes the model has already learned robust multilingual semantic structure.

Safety context distillation embeds context during pre-training: what counts as abusive targeting, what counts as protected identity, and what constitutes legitimate discussion of sensitive topics in a locale. This approach achieves 78–89% harm reduction across languages, far exceeding post-hoc RLHF for low-resource settings (Üstün et al., 2024). The Aya 23 models extended this with improved multilingual preference training (Aryabumi et al., 2024). Model merging (Aakanksha et al., 2024a) offers a complementary approach, combining safety-enhanced models with capability-focused models. Command A from Cohere exemplifies production deployment of these merged approaches (Cohere, 2025). While resource intensive and costly, these interventions are unavoidable: safety gaps persist when multilingual inclusion is treated as a downstream add-on rather than a first-order design requirement (Cohere, 2025).

Our framework does not require proportional low-resource representation in general pre-training. Safety Context Distillation and model merging target purposively constructed safety corpora, orders of magnitude smaller than general pre-training data, fully decoupling safety conditioning from capability training. We scope this framework intervention in tiers: near-term across the ∼20 languages with substantial deployment footprints; medium-term extending to informal-register varieties that formal web metrics systematically undercount (DataReportal et al., 2025; Fourreau, 2026).

Defense mechanisms show promise but require multilingual extension. Multi-round Automatic Red-Teaming (MART) reduces violation rates up to 84.7% while maintaining helpfulness (Chao et al., 2024). Robust Prompt Optimization achieves near-0% attack success through minimax defensive objectives (Agarwal et al., 2024). Prompt Adversarial Tuning (PAT) reduces attack success to nearly 0% while maintaining utility (Jain et al., 2024). However, these defenses remain primarily English-focused; extending them requires systematic multilingual red-teaming.

Multimodal expansion is critical. The Aya Vision project demonstrates vision-language models covering 23 languages can expand capabilities to languages spoken by over half the world's population (Dash et al., 2025). The M5 benchmark covering 41 languages reveals that larger models do not necessarily outperform smaller ones in multilingual multimodal settings (Kim et al., 2024), motivating targeted investment.

## 4.2. Participatory Design

This approach operationalizes what Mohamed et al. (2020) term *reverse tutelage*: centering expertise of historically marginalized communities. The Aya red-teaming dataset (Aakanksha et al., 2024b) exemplifies this by distinguishing "global" harms from "local" harms, a distinction English-only frameworks cannot make. Participatory design work with children and educators in Nigeria by Adejoro et al. (2023); Onimisi Adejoro et al. (2026) reveals local concerns such as access to electricity and agricultural sustainability that shape how AI harms are understood.

Participatory processes also improve technical outcomes by surfacing local euphemisms and dialectal variants likely to evade English-trained detectors, and identifying benign topics frequently over-refused. Gordon et al. (2023) demonstrate that crowdsourced moderation with diverse perspectives can ameliorate biases. *Participation includes the right to refuse*: indigenous communities may have legitimate concerns about cultural appropriation or surveillance (Bird, 2020). Participatory design must include meaningful consent mechanisms. This principle has practical implications: data collection initiatives should implement tiered consent (allowing communities to permit research use while prohibiting commercial deployment), establish benefit-sharing agreements, and create mechanisms for ongoing governance rather than one-time consent. The Te Hiku Media precedent in Aotearoa New Zealand, where Māori communities retained sovereignty over language data, offers a model (Bird, 2020).

## 4.3. New Metrics

Multilingual safety needs evaluation reflecting both curses. We propose that multilingual guardrails be evaluated on two headline metrics, reported jointly by language and locale: (1) Attack Success Rate (ASR): the probability that a model produces policy-violating content under adversarial prompting (Yong et al., 2023b; Zou et al., 2023); and (2) False Refusal Rate (FRR): the probability that a model refuses a benign request that a community-defined policy deems legitimate.

Both metrics should be computed under matched prompts across languages and audited with human review from fluent speakers. Language selection should use stratified sampling

across Joshi et al. (2020) resource tiers (0–5), targeting at minimum two languages per tier with geographic and script diversity, totaling approximately 20–25 languages. FRR is not only a usability metric; it is a *fairness* metric and a *governance* metric (high FRR enables "soft" censorship). Zheng et al. (2023) establish methodology for LLM-as-judge evaluation, but as Gureja et al. (2025) demonstrate, these judges are less reliable for low-resource languages, requiring human validation by fluent speakers as a technical necessity rather than a methodological preference.

We also recommend publishing disparity indices: absolute gaps between English and each target language, and worst-case gaps across the language set. If a system is marketed as multilingual, its safety claims should be evaluated as a distributional promise. Where thresholds cannot be met, model cards should clearly state the limitation. It is not currently standard practice for AI developers to list languages supported by an LLM (Peppin et al., 2025); mandating such transparency would enable fairer comparisons.

### 4.4. Call to Action

We conclude with specific recommendations for different stakeholders:

**ML Researchers:** (a) Develop safety benchmarks that include low-resource languages from the outset, building on efforts like MULTITuDE (Macko et al., 2023) and M5 (Kim et al., 2024); (b) Report disaggregated safety metrics by language rather than multilingual averages; (c) Investigate pre-training interventions that build safety into representations; (d) Evaluate defenses (MART (Chao et al., 2024), PAT (Jain et al., 2024), Robust Prompt Optimization (Agarwal et al., 2024)) across the linguistic spectrum; and (e) Extend multimodal safety research to low-resource contexts (Dash et al., 2025).

**Industry:** (a) Treat multilingual safety as a core product requirement, not an optional add-on; (b) Invest in annotation infrastructure and rater pools for underrepresented languages; (c) Implement Safety Context Distillation during pre-training (Üstün et al., 2024); (d) Publish language-specific ASR and FRR metrics and invite independent auditing; and (e) Disclose which languages are supported and to what degree (Peppin et al., 2025).

**Regulators:** (a) Mandate language-disaggregated safety reporting for AI systems deployed at scale; (b) Support multilingual dataset creation and local annotation capacity; (c) Recognise that safety disparities across languages constitute a fairness concern; and (d) Require transparency in language coverage claims.

**Civil Society:** (a) Document safety failures in local languages and advocate for community involvement; (b) Develop independent multilingual auditing capacity; (c) Ad-

vocate for FRR as a regulated metric alongside ASR; and (d) Support participatory initiatives like Masakhane, NusaCrowd, and Aya.

## 5. Conclusion

This paper argues that multilingual AI safety currently imposes a Dual Curse on low-resource language users: heightened vulnerability to harmful outputs and reduced access to benign assistance. The evidence suggests this pattern is reinforced by the big head focus of safety literature, by translation attack pathways bypassing English filters, and by culturally misaligned refusal policies. Critically, we identify the pre-training bottleneck as a primary structural mechanism: reward models achieve only random-chance accuracy for low-resource languages, rendering post-hoc alignment structurally ineffective.

These failures create governance risks when automated moderation is mandated at national scale, including in at least 22 countries. The 7 million appeals to Meta's Oversight Board in a single month, the 38% conflicting predictions in content moderation systems, and systematic over-removal affecting Global South users all point to a system that is simultaneously porous to harm and restrictive of legitimate expression.

The path forward is a set of socio-technical guardrails: earlier safety conditioning during pre-training (78–89% harm reduction), participatory mechanisms letting communities specify what harm looks like, and evaluation regimes treating false refusals as seriously as jailbreak success. The vicious cycle identified by Peppin et al. (2025) demonstrates why passive approaches are insufficient.

If the ML community continues to export English-optimized safety stacks without local accountability, it risks reproducing digital coloniality: historically marginalized communities receive worse protection and worse service, while their languages are weaponized as attack vectors. If, instead, the field commits to socio-technical equity, multilingual safety can become a driver of inclusion: safer models that are also more helpful, in more languages, for more people.

Global AI systems are only as safe as their weakest linguistic channels, and the long tail is where both human vulnerability and adversarial opportunity concentrate. Breaking the Dual Curse requires recognizing that safety is not a universal property to be assumed but a socio-technical achievement to be built, language by language, community by community. The long tail is not a technical afterthought; it is the domain through which 5 billion people communicate. Their safety cannot wait for scaling laws to eventually reach them.

## Impact Statement

This paper is centrally concerned with broader societal impacts: it documents, diagnoses, and proposes remedies for an equity failure in AI safety that disproportionately harm speakers of low-resource languages. We argue for community-engaged harm specification, language-disaggregated safety reporting, and pre-training interventions that build safety equitably rather than retrofit it onto Anglophone foundations. Adoption of this framework would shift accountability for safety claims from aggregate multilingual averages toward worst-case distributional guarantees, benefiting populations currently underprotected by deployed systems.

We acknowledge potential risks of this work. The paper synthesizes published evidence of cross-lingual attack pathways (translation attacks, multimodal exploits, universal adversarial suffixes) and documents governance hazards including the use of automated moderation infrastructure for censorship in at least 22 countries. We introduce no new attack techniques and cite only findings already in the public literature. Several headline quantitative claims, notably the 35-fold harmfulness gap and 49–50% reward model accuracy, rest primarily on a single source and warrant independent replication across a broader language set before generalization. Our systematic review is restricted to English-language publications, a constraint that is itself a self-exemplifying instance of the bias we document. Finally, our framework foregrounds affected communities' right to define harm and right to refuse participation; we caution against operationalizations that extract community labor without benefit-sharing or treat consent as a one-time occurrence rather than ongoing.

## Acknowledgements

This work was supported in part by U.S. NSF awards #2114824 and #2438810. DISTRIBUTION STATEMENT A. Approved for public release. Distribution is unlimited. This material is based upon work supported by the Department of the Air Force under Air Force Contract No. FA8702-15-D-0001 or FA8702-25-D-B002. Any opinions, findings, conclusions or recommendations expressed in this material are those of the author(s) and do not necessarily reflect the views of the Department of the Air Force. © 2026 Massachusetts Institute of Technology. Delivered to the U.S. Government with Unlimited Rights, as defined in DFARS Part 252.227-7013 or 7014 (Feb 2014). Notwithstanding any copyright notice, U.S. Government rights in this work are defined by DFARS 252.227-7013 or DFARS 252.227-7014 as detailed above. Use of this work other than as specifically authorized by the U.S. Government may violate any copyrights that exist in this work.

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

# A. Dialectal Ablation: Per-Dialect Results and Framework Illustrations

This appendix provides supporting visuals for the empirical ablation reported in Section 2 and the framework proposed in Section 4. Table 1 gives per-dialect zero-shot results for the safety-guard evaluation across 50 English dialects. Figure 4 illustrates the contrast between current post-hoc RLHF practice and the proposed Safety Context Distillation and model-merging interventions. Figure 5 maps each stage of the three-stage participatory pipeline to its anchoring existence proof.

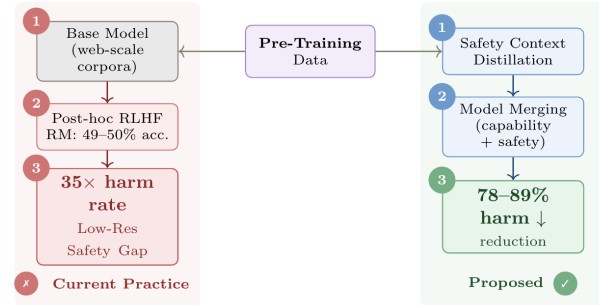

*Figure 4.* Pre-training intervention contrasts current post-hoc RLHF practice, where reward model accuracy falls to random chance for low-resource languages, with proposed Safety Context Distillation and model merging achieving 78–89% harm reduction.

# B. Glossary of Key Terms

**Harmful content** refers to material that causes or risks causing injury, including hate speech targeting protected characteristics (race, ethnicity, religion, gender, sexuality), instructions for violence or illegal activities, and content designed to deceive or manipulate (Weidinger et al., 2022; Shelby et al., 2023). What constitutes harm varies across

| Dialect / Reference | Acc (%) | Δ vs SAE |
|---|---|---|
| Standard American English (SAE) | 82.8 | – |
| Urban AAVE | 68.2 | −14.6 |
| Rural AAVE | 67.4 | −15.4 |
| Chicano English | 68.2 | −14.6 |
| Jamaican English | 67.4 | −15.4 |
| Nigerian English | 68.1 | −14.7 |
| Pure Fiji English | 66.8 | −16.0 |
| 50-dialect mean | 67.1 | −15.7 |
| White Zimbabwean Eng. (worst) | 55.6 | −27.2 |

*Table 1.* Zero-shot harmful content classification accuracy across 50 English dialects using Qwen3Guard-8B, the strongest zero-shot performer among 20 evaluated safety guards. The six dual-bind dialects exhibit substantial gaps in demographically salient speaker populations; White Zimbabwean English carries the largest raw drop, indicating that the underlying mechanism is structural distance from the training distribution rather than demographic targeting alone.

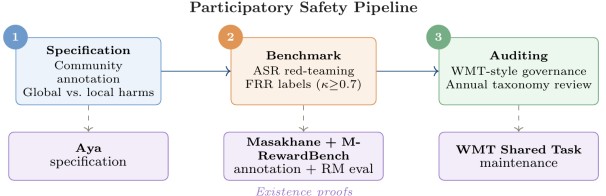

*Figure 5.* Three-stage participatory safety pipeline mapping each stage to anchoring existence proofs: Aya (specification and red-teaming), Masakhane and M-RewardBench (annotation governance and RM evaluation), and WMT shared task (maintenance model).

| Technique | High-Res | Low-Res | Gap |
|---|---|---|---|
| CHAT-RLHF | 44.8% | 23.4% | 1.9× |
| xSFT | 23.0% | 9.8% | 2.3× |
| xRLHF | 14.4% | **2.4%** | 6.0× |
| Safety Distillation | 85% | 78% | 1.1× |
| RM Accuracy | 63–66% | 49–50% | – |

*Table 2.* Alignment effectiveness by language resource level (harm reduction in percentage points). CHAT-RLHF refers to the official LLaMA-2-chat checkpoint; xSFT and xRLHF refer to the authors' multilingual supervised fine-tuning and reward model training. Post-hoc methods show widening gaps as alignment sophistication increases; pre-training Safety Distillation achieves near-parity. Data from Shen et al. (2024); Üstün et al. (2024).

| Metric | Value | Source |
|---|---|---|
| Languages globally | 7,164 | (Eberhard et al., 2024) |
| Languages w/ NLP data | ∼1,500 | (Peppin et al., 2025; Joshi et al., 2020; Yong et al., 2023a) |
| GPT-3 English tokens | 92.65% | (Zhao et al., 2024) |
| High-res harmful rate | ∼1% | (Shen et al., 2024) |
| Low-res harmful rate | ∼35% | (Shen et al., 2024) |
| **Harmfulness gap** | **35×** | (Shen et al., 2024; Yong et al., 2023b) |
| Low-res RM accuracy | 49–50% | (Shen et al., 2024) |
| Tokenization cost gap | 5.6× | (Ahia et al., 2023) |
| Countries w/ auto-mod | 22+ | (Freedom House, 2023) |

*Table 3.* Key metrics documenting the multilingual safety divide (RM: reward model).

contexts: caste-based slurs in South Asia, ethnic targeting in East Africa, or sectarian incitement in the Middle East represent locally-specific harms that English-trained systems often fail to recognize (Aakanksha et al., 2024b).

**Toxic outputs** are model generations containing harmful content, including slurs, threats, dehumanizing language, or material promoting violence against individuals or groups. Toxicity classifiers attempt to detect such outputs, but their accuracy varies dramatically across languages (Shen et al., 2024).

**Safety mechanisms** are technical systems designed to prevent harmful outputs, including toxicity classifiers, content filters, refusal policies, and alignment training (Ouyang et al., 2022; Bai et al., 2022). These mechanisms are often described as "guardrails."

**Universal adversarial attacks** are optimized input sequences (prompts or suffixes) that reliably induce aligned models to violate their safety policies across diverse queries (Zou et al., 2023). Unlike manually crafted jailbreaks, these attacks are algorithmically generated and can transfer across models, making them particularly dangerous when combined with cross-lingual translation.

**Automated moderation** refers to the use of machine learning systems to detect, flag, or remove content at scale, typically deployed by platforms and increasingly mandated by governments (Gorwa et al., 2020; Gillespie, 2020).

**Low-resource languages** are languages with limited digitized text, constrained annotation infrastructure, and sparse representation in AI training data, not necessarily languages with few speakers (Joshi et al., 2020; Ranathunga & de Silva, 2022). Javanese (83 million speakers) and Hausa (77 million speakers) qualify as low-resource due to limited digital presence.

**Epistemic hegemony** refers to the systematic dominance of one knowledge system over others, where the standards, methods, and categories of the dominant group become naturalized as universal while alternative ways of knowing are marginalized or delegitimized (Grosfoguel, 2013).

**Epistemic plurality** refers to the recognition and validation of multiple knowledge systems as legitimate ways of understanding the world, rather than privileging a single (typically Western) epistemological framework (de Sousa Santos, 2014).

**Algorithmic coloniality** refers to the ways AI systems reproduce colonial power hierarchies by embedding the assumptions, values, and knowledge structures of dominant (typically Western, English-speaking) cultures while marginalizing others (Mohamed et al., 2020). This includes training data imbalances, evaluation metrics derived from Western legal frameworks, and annotation practices that

treat Anglophone norms as universal.

**Digital coloniality** describes a contemporary form of domination where technological infrastructure, data extraction, and algorithmic systems extend the economic and epistemic control historically exercised through territorial colonialism (Kwet, 2019). In AI safety, this manifests when English-optimized systems are deployed globally without accountability to local communities.

**Decolonization in AI** is the process of dismantling colonial power structures embedded in AI systems through: (1) critical examination of how technical choices reproduce inequities; (2) centering the expertise of historically marginalized communities (what Mohamed et al. (2020) call "reverse tutelage"); and (3) building coalitions that advocate for equitable AI development. Decolonization cannot be achieved through methods aimed at inclusion alone, but rather requires the fundamental restructuring of authorial power, i.e., who defines problems, solutions, and successes.

## C. PRISMA Methodology

### C.1. Search Strategy

We conducted systematic searches across two academic databases and supplemented with venue-specific searches:

**Database Searches:**

- Scopus: 1,941 documents

- Web of Science: 774 documents

- Combined total: 2,715 documents

**Venue-Specific Searches:** Systematic searches across ACL, EMNLP, NAACL, NeurIPS, ICML, ICLR, FAccT, CHI, ICWSM, WWW, and arXiv yielded an additional 385 documents.

**Search Terms:** Queries combined terms for large language models ("LLM," "large language model," "GPT," "transformer") with terms for safety and alignment ("safety," "alignment," "toxicity," "hate speech," "harmful content," "jailbreak," "guardrail," "refusal") and multilinguality ("multilingual," "cross-lingual," "low-resource," "non-English," "Global South").

**Complete Search Strings:**

*Scopus (Primary): TITLE-ABS-KEY(("large language model" OR "LLM" OR "GPT" OR "transformer" OR "foundation model" OR "ChatGPT" OR "Claude" OR "Llama" OR "generative AI") AND ("safety" OR "alignment" OR "toxicity" OR "hate speech" OR "harmful" OR "jailbreak" OR "guardrail" OR "refusal" OR "red-team" OR "adversarial" OR "content moderation") AND ("multi-*

*lingual" OR "cross-lingual" OR "low-resource" OR "non-English" OR "Global South" OR "African language" OR "translation attack")) AND PUBYEAR ¿ 2017*

*Web of Science: TS=(("large language model*" OR "LLM" OR "GPT*" OR "transformer*" OR "foundation model*") AND (safety OR alignment OR toxic* OR "hate speech" OR harmful OR jailbreak* OR guardrail*) AND (multilingual OR "cross-lingual" OR "low-resource" OR "non-English")); Timespan: 2018–2025*

*ACL Anthology: (multilingual OR cross-lingual OR low-resource) AND (safety OR toxicity OR hate speech OR jailbreak OR harmful OR alignment); Venues: ACL, EMNLP, NAACL, Findings, AACL, EACL, CoNLL*

*arXiv (cs.CL, cs.AI, cs.LG): (multilingual OR cross-lingual) AND (LLM OR "large language model") AND (safety OR jailbreak OR toxicity OR alignment)*

**Date Range:** January 2018 – October 2025 (covering the transformer era)

**Language:** English-language publications only (due to reviewer language constraints, acknowledging this as a limitation that itself reflects the English bias we document)

### C.2. PRISMA Flow Diagram

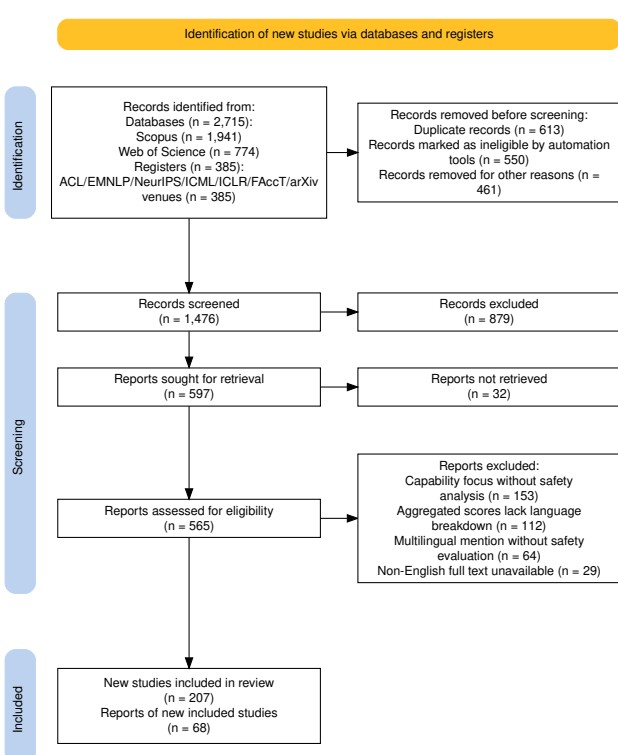

*Figure 6.* PRISMA 2020 flow diagram for the systematic review of multilingual AI safety studies.

## C.3. Inclusion and Exclusion Criteria

**Inclusion criteria:**

- Multilingual safety evaluation with disaggregated results by language

- Explicit analysis of cross-lingual attacks or safety failures

- Multilingual safety benchmarking with language-specific metrics

- Governance analysis tied to multilingual moderation and alignment

- Large-scale collaborative initiatives with substantial empirical findings on multilingual safety

- Studies examining translation-based attacks or defenses

- Research on culturally-specific harm categories across languages

**Exclusion criteria:**

- Studies mentioning "multilingual" only in model descriptions without safety analysis

- Studies reporting only aggregate multilingual scores without language disaggregation

- Non-peer-reviewed sources (except high-impact arXiv preprints or major initiative reports)

- Studies focused solely on capability (e.g., translation quality) without safety implications

- Duplicate publications or extended abstracts of included full papers

- Studies published before January 2018 (pre-transformer era)

## C.4. Data Extraction

For each included study, we extracted:

- Languages evaluated (categorized by resource level)

- Safety metrics reported (ASR, toxicity rate, refusal rate, etc.)

- Model(s) evaluated

- Attack types examined (if applicable)

- Key findings on cross-lingual safety disparities

- Proposed mitigations or frameworks

## C.5. Limitations

It is worth acknowledging the limitations of any review in this area. Industry safety work is often reported selectively, and some evaluations are shared only in model cards or system descriptions rather than in archival publications. Safety benchmarks evolve rapidly, and some recent work may not yet appear in indexed databases. Despite these limitations, the pattern is clear enough to support a robust claim: the multilingual safety literature remains small relative to monolingual English safety, and it is heavily skewed toward a small set of globally dominant languages.

Two structural limitations warrant additional emphasis. First, our search is restricted to English-language publications, a standard but consequential constraint that is itself a self-exemplifying instance of the English-dominance problem we document. Non-English safety scholarship, where it exists, is underindexed in Scopus and Web of Science; the 8.6% low-resource coverage figure therefore likely overstates true breadth. Second, both this literature and the training corpora it evaluates are anchored in formally produced web content (news organizations, government portals, commercial publishers). Contemporary web metrics (DataReportal, October 2025) show 20 languages covering approximately 97% of websites by traffic share, but this measures institutional content production rather than actual user language distribution. Languages like Hausa (77M speakers) and Javanese (83M speakers) are used at enormous scale in social media, messaging platforms, and informal digital settings that formal web crawls systematically exclude, precisely the registers where the Harmfulness Curse is most acute and English-trained classifiers are least reliable. A safety research program tethered to the formal web will perpetually underestimate both the scale of the problem and the populations most at risk. Several headline claims, notably the $35\times$ harmfulness gap and 49–50% RM accuracy, rest primarily on Shen et al. (2024) and require independent replication across a broader language set before generalizing beyond their 19-language evaluation.

## C.6. Quality Assessment

We assessed study quality using criteria adapted for AI safety research:

**Methodological rigor:**

- Clear description of models evaluated

- Reproducible experimental setup (code/data availability)

- Appropriate statistical analysis or confidence intervals

- Language-disaggregated reporting

**Validity of findings:**

- Sample size adequate for claimed conclusions

- Appropriate baselines and comparisons

- Acknowledgment of limitations

- External validity considerations (generalization beyond tested languages)

**Quality ratings:** Studies were rated as High (meets all criteria), Medium (meets most criteria with minor gaps), or Low (significant methodological concerns). Of 207 included studies: 89 (43%) rated High, 94 (45%) rated Medium, 24 (12%) rated Low. Low-rated studies were retained when they provided unique evidence on understudied languages or attack vectors.

### C.7. Risk of Bias Assessment

We identified several sources of potential bias in the literature:

**Publication bias:** Positive results (successful attacks, significant disparities) are more likely to be published than null findings. This may inflate estimates of vulnerability.

**Language selection bias:** Researchers tend to evaluate languages they speak or have resources for, creating coverage gaps. The 18 studies evaluating low-resource languages disproportionately focused on languages with academic communities (e.g., Swahili, Yoruba) rather than those with no research infrastructure.

**Model selection bias:** Most studies evaluate commercially prominent models (GPT-4, Claude, LLaMA), with limited coverage of regional models (e.g., Chinese LLMs, Arabic-focused models).

**Temporal bias:** The field evolves rapidly; findings from 2023 may not reflect 2025 model capabilities. We weighted recent studies more heavily in synthesis.

**Industry reporting bias:** Companies selectively report safety evaluations, often omitting low-resource language results or aggregating across languages to obscure disparities.

## D. Theoretical Foundations

### D.1. Socio-Materiality

Socio-materiality theory (Orlikowski, 2010) provides a framework for understanding how material conditions and social practices are mutually constitutive: they "imbricate" to produce sociotechnical systems. In the context of multilingual AI safety, this means recognizing that:

1. **Material constraints shape capabilities:** The availability of digitized text, annotation infrastructure, and compute resources directly determines which languages receive safety attention. These are not neutral technical constraints but reflect historical patterns of investment, colonialism, and economic development.

2. **Technical artifacts encode social relations:** When a model is trained on 92.65% English data, it does not merely lack coverage of other languages; it actively embeds English-centric assumptions about harm, offense, and legitimate speech into its representations.

3. **Imbrication of human and material agency:** Safety outcomes emerge from the entanglement of human decisions (which languages to prioritize, which harm categories to define) and material affordances (data availability, model architecture, evaluation infrastructure).

The Aya Initiative's documentation of real-world barriers (power outages affecting contributors, civil conflicts in Myanmar and Armenia limiting participation, absence of postal infrastructure in Somalia and Yemen preventing device distribution) illustrates how material conditions constrain who can participate in defining AI safety (Singh et al., 2024; Peppin et al., 2025). Following Gibson & Carmichael (1966) and Norman (1988), we can understand these as *affordances*: opportunities and constraints that emerge from the interaction of technological capabilities and social contexts.

Cognitive load theory (Sweller, 2011) provides additional insight: humans' short-term memory has finite capacity that becomes overloaded when presented with excessive information. In multilingual settings experiencing harmful content proliferation, this cognitive overload is exacerbated by the need to process information across languages, modalities, and cultural contexts, making it difficult to discern reliable information and causing reluctance to accept legitimate interventions.

### D.2. Decolonial AI

Mohamed et al. (2020) propose three tactics for addressing algorithmic coloniality, each directly relevant to the Dual Curse:

**1. Critical Technical Practice:** Examining how technical choices (training data composition, evaluation benchmarks, annotation guidelines) reproduce colonial power relations. The 92.65% English dominance in GPT-3 training data is not a neutral technical fact but a reproduction of global knowledge hierarchies that privileges certain epistemologies while marginalizing others.

**2. Reverse Tutelage:** Centering the expertise of communities in the Global South rather than treating them as passive recipients of safety technologies developed elsewhere. Participatory design initiatives like Aya and Masakhane exemplify this approach, demonstrating that affected communities can and should shape how harm is defined and measured in their linguistic contexts.

**3. Renewal of Political Communities:** Building coalitions that can advocate for equitable AI development. This includes supporting local annotation capacity, funding multilingual safety research led by affected communities, and creating accountability mechanisms for global AI deployments.

The Dual Curse validates these arguments empirically: colonial knowledge hierarchies manifest in the 92.65% English training concentration; epistemic injustice appears in the 20-point relevance gap that denies communities the ability to interact with AI systems in their own languages; and the need for material reparations (not merely procedural fixes) is evidenced by the failure of post-hoc alignment to remediate pre-training inadequacies. Gwagwa et al. (2024) emphasize that decolonization requires recognizing persistent colonial repercussions leading to biases in AI solutions and disparities based on gender, race, geography, and income. The relevance curse, where benign queries in low-resource languages are refused while equivalent English queries are answered, can be understood as a form of epistemic violence: the systematic devaluation of local knowledge and the imposition of Anglophone norms as universal standards.

Birhane et al. (2022) provide empirical support, finding in their annotation of 526 papers from FAccT and AIES (2018–2021) that 40% of papers were "agnostic" to disparate impacts on marginalized groups, with most papers using implicit rather than explicit mentions of impacted communities. This emphasizes the need for AI ethics grounded in concrete use-cases, people's experiences, and approaches sensitive to structural and historical power asymmetries.

### D.3. Historical Context of Linguistic Colonialism

AI systems risk reproducing the epistemic hegemony of Western modernity instituted during the colonial period (Quijano, 2000; Mignolo, 2011). From 1492 onwards, colonial powers systematically depreciated linguistic diversity through criminalizing non-European and indigenous languages and enforcing linguistic conversion, thereby delegitimizing non-European knowledge systems: cosmologies, biological sciences, medical knowledge, and cultural production (Bhola, 1987; Mignolo, 2003). This systematic privileging of Eurocentric languages contributed to what de Sousa Santos (2014) terms "epistemicide," the suppression of alternative ways of knowing that has shaped global knowledge hierarchies to the present day.

Contemporary AI development echoes this pattern. When 92.65% of GPT-3's training tokens are English, when safety benchmarks evaluate harm through Western legal and corporate categories, and when annotation guidelines encode Anglophone cultural assumptions, AI systems perpetuate rather than challenge epistemic hierarchies. The Dual Curse is thus not merely a technical failure but a reproduction of colonial knowledge structures in algorithmic form.

### D.4. Abstraction Traps

Selbst et al. (2019) identify five "abstraction traps" that occur when technical solutions are developed without attention to social context. Four are particularly relevant:

**The Portability Trap:** Assuming that a technical solution developed in one context will work in another. English safety guardrails are not portable to other languages because they encode culturally specific assumptions about harm. The 35-fold harmfulness gap demonstrates the consequences of this assumption.

**The Solutionism Trap:** Believing that technical fixes can address fundamentally social problems. The pre-training bottleneck cannot be solved by better alignment algorithms alone; it requires material investment in multilingual data, annotation infrastructure, and community engagement. The near-zero effectiveness of xRLHF (2.4% for low-resource languages) illustrates why algorithmic improvements without upstream investment cannot succeed.

**The Framing Trap:** Failure to model the entire system in which a technology operates. Content moderation systems that focus only on detecting harmful content without considering false refusal rates, user trust, and governance implications miss critical dimensions of the problem.

**The Ripple Effect Trap:** Failure to understand how technical interventions change the social context. Automated moderation normalizes certain forms of speech suppression, making it easier to expand the scope of "harm" to include dissent, minority advocacy, or investigative journalism.

### D.5. Foundation Model Risks

Bommasani et al. (2021) provide foundational analysis of the opportunities and risks of foundation models, noting that their scale and generality create both unprecedented capabilities and novel failure modes. The homogenization risk (where a single model's failures propagate across all downstream applications) is particularly relevant to multilingual safety. When 92.65% of training data is English, the resulting model's cultural assumptions about harm become embedded in every application built on that foundation.

# E. Cross-Domain Convergence with the Bias-and-Fairness Literature

The argument advanced in the main text is corroborated, point by point, by the most comprehensive recent treatment of bias and fairness in LLMs: Gallegos et al. (2024), published in *Computational Linguistics*. Despite emerging from a research tradition focused primarily on English-language gender and racial stereotyping, the survey reaches conclusions that align closely with the multilingual safety diagnosis presented here. This appendix documents these parallels.

## E.1. Convergence on English Dominance as an Open Problem

Gallegos et al. (2024) explicitly name "expanding language resources" beyond English as an unresolved research challenge, and acknowledge that their own survey is restricted to English-language works. That the field's most comprehensive bias survey reproduces the very language imbalance it studies underscores the structural depth of the problem. Two independent research communities, multilingual NLP safety and English-focused bias-and-fairness, converge on the same diagnosis: English dominance is an unresolved structural problem rather than a transitional artifact.

## E.2. Convergence on the Solutionism Trap

Gallegos et al. (2024) catalog over 200 bias mitigation techniques operating at every stage of the LLM pipeline: counterfactual data augmentation and demographic reweighting at pre-processing, RLHF and Constitutional AI during training, decoding constraints and weight redistribution at inference, and output rewriting post-hoc. Despite this taxonomic breadth, they conclude that "technical solutions are incomplete without broader societal action against power hierarchies that diminish and dominate marginalized groups." This conclusion mirrors Selbst et al. (2019)'s solutionism trap. Critically for our argument, their analysis reveals weak cross-stage generalization even for English-focused techniques: biases mitigated in the embedding space re-emerge in downstream tasks, and reward signals derived from crowd-sourced annotations encode culturally specific norms from dominant groups. For low-resource languages, these weaknesses compound: SFT data is not merely sparse but likely annotated through Anglophone harm taxonomies, meaning the reward model's training signal is doubly compromised. This is why no downstream technique in their full pipeline taxonomy can recover the gradient signal a chance-accuracy RM has lost.

## E.3. Convergence on Disparate System Performance

Gallegos et al. (2024) formalize dialect-level and cross-linguistic performance gaps under the category *disparate system performance*, defined as degraded understanding, diversity, or richness in language processing or generation between social groups or linguistic variations. Their survey documents that this harm type is systematically underaddressed in the mitigation literature, which overwhelmingly targets gender and racial stereotyping in English rather than cross-linguistic or dialectal variation. The Relevance Curse described in our main text is an instance of disparate system performance at scale, lacking the remediation infrastructure that other bias types have attracted.

## E.4. Convergence on Benchmark Critique

Gallegos et al. (2024) caution that benchmarks steering model development tend to express the perspectives of dominant groups in the name of objectivity, and that "universal" evaluation frameworks risk perpetuating harm against marginalized communities. They further identify disaggregated, group-specific reporting as a best practice the field has consistently failed to adopt, with aggregate scores obscuring disparate performance across groups. Our recommendation that multilingual safety claims be evaluated as distributional promises (with disparity indices across languages) addresses precisely this failure mode.

## E.5. Convergence on the Participatory Turn

Gallegos et al. (2024) identify "developing participatory research designs" as a priority open challenge for bias mitigation, citing community-in-the-loop frameworks as essential for representative harm specification across social groups. That two research traditions reach the same prescription strengthens the normative case for the three-stage participatory pipeline proposed in Section 4: participatory harm specification is not a culturally specific preference but a technical necessity recognized across the field.

## E.6. Convergence on Automated Evaluation Reliability

Gallegos et al. (2024) document that bias evaluation metrics based on automated classifiers are unreliable even for English: toxicity classifiers disproportionately flag African American English, and sentiment classifiers misclassify statements about stigmatized groups. This finding complements Gureja et al. (2025)'s low-resource judge degradation, jointly supporting our argument that human validation by fluent speakers is a technical requirement rather than a methodological preference.

# F. Extended Evidence Tables

This appendix provides extended evidence tables that support the main arguments presented in the paper. Table 4 synthesizes the Dual Curse framework, mapping symptoms to mechanisms and guardrail implications. Tables 5–9 provide

detailed breakdowns of pre-training data distributions, alignment effectiveness, attack techniques, language resource classifications, and key participatory initiatives.

## G. Governance and Policy Implications

The Dual Curse has significant governance implications beyond technical performance metrics. This appendix documents real-world cases where multilingual safety failures contributed to societal harms, surveys the evolving regulatory landscape, and presents data on internet shutdowns that compound AI safety challenges.

### G.1. Documented Governance Failures

Table 10 summarizes documented cases where inadequate multilingual content moderation contributed to real-world harms. These cases demonstrate that the Dual Curse is not merely an academic concern but has material consequences for human safety and democratic processes.

### G.2. Regulatory Landscape

Table 11 surveys major regulatory frameworks addressing AI content moderation and their provisions (or lack thereof) for multilingual contexts. A consistent pattern emerges: regulations acknowledge linguistic diversity implicitly but rarely mandate language-disaggregated safety reporting or evaluation.

### G.3. Internet Shutdown Data

Access Now's #KeepItOn coalition documented 283 internet shutdowns across 39 countries in 2023 (Rosson et al., 2024). These shutdowns disproportionately affect regions where low-resource languages predominate, creating information vacuums that AI systems cannot address even if they were safe:

- **India**: 116 shutdowns (highest globally), affecting Hindi, Kashmiri, Punjabi speakers

- **Myanmar**: 14 documented shutdowns during ongoing conflict, affecting Burmese, Karen, Shan

- **Iran**: 8 shutdowns during protests, affecting Farsi, Kurdish, Azerbaijani

- **Ethiopia**: 4 shutdowns during conflict, affecting Amharic, Tigrinya, Oromo

- **Pakistan**: 16 shutdowns, affecting Urdu, Punjabi, Sindhi, Pashto

The intersection of internet shutdowns and AI safety gaps creates compounded vulnerability: when connectivity is restored, users face AI systems that are simultaneously more dangerous (harmfulness curse) and less helpful (relevance curse) in their languages.

## H. Defense Mechanisms: Detailed Analysis

Table 12 compares the effectiveness of various defense mechanisms against jailbreaking attacks, highlighting a critical gap: most defenses have been evaluated only in English, leaving their multilingual effectiveness unknown. Only Safety Context Distillation has demonstrated cross-lingual effectiveness, achieving 78% harm reduction even in low-resource languages.

## I. Bias and Arbitrariness: Extended Evidence

Table 13 documents specific bias patterns identified in AI content moderation systems. These biases compound the Dual Curse by systematically disadvantaging speakers of non-standard dialects and members of marginalized communities, who face both higher false positive rates (relevance curse) and reduced protection from actual harms (harmfulness curse).

## J. Multilingual Benchmarks and Datasets

Table 14 surveys key multilingual safety benchmarks and datasets. A notable pattern emerges: most benchmarks either focus on English, cover only high-resource languages, or prioritize capability over safety evaluation. This benchmark gap perpetuates the measurement bias that underlies the Dual Curse.

## K. Tokenization Disparities: Extended Data

Table 15 presents detailed tokenization cost data across scripts and language families. These disparities have direct implications for both cost (users pay more per query) and safety (longer token sequences consume more context, potentially degrading safety behavior). The 5.6× cost ratio for Tibetan versus English represents a concrete economic barrier to equitable AI access.

*Table 4.* The Dual Curse: Symptoms, Mechanisms, and Guardrail Implications

| Dimension | Symptoms | Mechanisms | Guardrail Implications |
|---|---|---|---|
| **Harmfulness Curse** | Higher harmful output rate; 50–80% translation attack success; $3\times$ vulnerability increase; 5.7K attack clusters identified | English-trained toxicity classifiers; reward model bias (49–50% = random); surface-cue dependence; morphological blindness; multimodal attack vectors | Pre-training safety corpora; multilingual reward models; local harm taxonomies; adversarial evaluation across languages; MART/PAT defenses |
| **Relevance Curse** | 20-point instruction-following drop; elevated false refusal rates; culturally thin responses; 38% conflicting predictions; dialect bias | Western policy operationalization; Anglophone rater pools; missing local context; narrow harm mappings; algorithmic arbitrariness | Participatory policy specification; community-defined legitimacy; local exemplar curation; false refusal as first-class metric; diverse annotator pools |
| **Primary Structural Mechanism:** | RM accuracy 49–50% (random); RLHF $\sim$0% improvement for low-resource; unreliable LLM-as-judge | Pre-training data deficit; RM cannot distinguish good/bad responses; vicious cycle of synthetic data bias | Pre-training data investment; Safety Context Distillation; dedicated multilingual RM training; model merging |
| **Weaponization** | 22+ countries mandating automated moderation; 7M Meta appeals; asymmetric accountability; 13% multilingual centrality | Safety tooling as governance infrastructure; opacity in low-resource contexts; weak auditing capacity; multilingual bridge exploitation | Transparency requirements; language-specific safety claims; independent auditing; disparity reporting; federated approaches |

*Table 5.* Pre-Training Data Distribution Across Major Models

| Model | English % | Other High-Res % | Low-Res % | Languages |
|---|---|---|---|---|
| GPT-3 | 92.65 | 6.35 | <1[†] | $\sim$100 |
| LLaMA-2 | 89.70 | 9.30 | <1[†] | $\sim$20 |
| LLaMA-3 | $\sim$85.00 | $\sim$13.00 | <2[‡] | $\sim$30 |
| Mistral-7B | $\sim$89.00 | $\sim$10.00 | <1[‡] | $\sim$10 |
| Command A | $\sim$70.00 | $\sim$25.00 | $\sim$5[‡] | $\sim$23 |
| BLOOM | 30.00 | 40.00 | 30.00 | 46 |
| Aya-101 | 15.00 | 35.00 | 50.00 | 101 |
| GPT-4o | N/A[§] | N/A[§] | N/A[§] | N/A[§] |
| Gemini Ultra | N/A[§] | N/A[§] | N/A[§] | N/A[§] |

[†] Exact percentages not disclosed; estimated from token-level analysis in Zhao et al. (2024).

[‡] Estimated from available technical reports (Dubey et al., 2024; Jiang et al., 2023; Cohere, 2025); exact figures not disclosed.

[§] Training data composition not publicly disclosed. Opacity in language-disaggregated

reporting by closed frontier models constitutes a governance concern, precluding independent safety auditing.

*Table 6.* Alignment Effectiveness: Detailed Breakdown by Technique and Language Resource Level

| Technique | High-Res | Med-Res | Low-Res | Gap Ratio | Source |
|---|---|---|---|---|---|
| RLHF | 45% | 32% | 20% | $2.25\times$ | Shen et al. |
| xSFT | 20% | 14% | 7% | $2.86\times$ | Shen et al. |
| xRLHF | 14% | 8% | 2.4% | $5.83\times$ | Shen et al. |
| Safety Distillation | 85% | 82% | 78% | $1.09\times$ | Üstün et al. |
| Model Merging | 83% | 80% | 76% | $1.09\times$ | Aakanksha et al. |
| **RM Accuracy** | 63–66% | 55–58% | 49–50% | – | Shen et al. |

*Table 7.* Attack Techniques and Their Multilingual Implications

| Attack Type | Description | Multilingual Risk | Source |
|---|---|---|---|
| Translation attacks | Translate harmful prompts to low-resource languages | 50–80% success | (Yong et al., 2023b) |
| Universal suffixes | Optimized suffixes transfer across models | Cross-model transfer | (Zou et al., 2023) |
| DeepInception | Nested scenarios bypass safety through "hypnosis" | Unexplored in LRL | (Yu et al., 2024) |
| Many-shot | Exploit long context windows (4,000+ tokens) | Amplified in LRL | (Anil et al., 2024) |
| Prompt injection | Manipulate translation systems | All language pairs | (Wang et al., 2024c) |
| Multimodal | Vision-language attack surfaces | 75% on GPT-4V | (Chen et al., 2024) |
| Audio-visual | Cross-lingual phonetic exploits | Emerging threat | (Xu et al., 2024) |
| Image-rendered text | Prompts as images bypass filters | Pronounced in LRL | (Derner & Batistič, 2025) |

*Table 8.* Language Resource Classification Used in This Review

| Resource Level | Characteristics | Example Languages | Studies (n=207) |
|---|---|---|---|
| English | Dominant in training, benchmarks, and safety research | English | 127 (61.4%) |
| High-Resource | Well-represented in training data; established NLP resources | Chinese, Spanish, French, German, Japanese, Arabic | 62 (30.0%) |
| Medium-Resource | Some digital presence; limited safety evaluation | Hindi, Indonesian, Vietnamese, Thai, Turkish | 12 (5.8%) |
| Low-Resource | Sparse training data; minimal safety research | Hausa, Igbo, Javanese, Swahili, Zulu | 5 (2.4%) |
| Very Low-Resource | Extremely limited data; virtually no safety coverage | Kamba, Scots Gaelic, many indigenous languages | 1 (0.5%) |

*Table 9.* Key Participatory Initiatives for Multilingual AI

| Initiative | Scope | Contribution | Safety Relevance |
|---|---|---|---|
| Aya | 101 languages, 119 countries, 3,000+ contributors | Instruction-tuned models; red-teaming dataset | Global/local harm distinction; safety distillation |
| Masakhane | African languages | MT benchmarks; participatory research model | Community-defined evaluation; local capacity |
| NusaCrowd | Indonesian languages (700+) | Unified data hub; multimodal resources | Regional harm categories; dialect coverage |
| SEACrowd | Southeast Asian languages | Multilingual multimodal benchmark | Cross-lingual evaluation standards |

*Table 10.* Documented Cases of Multilingual Safety Failures with Governance Implications

| Context | Failure Type | Evidence | Governance Implication |
|---|---|---|---|
| Myanmar (2018–present) | Under-moderation of hate speech in Burmese | UN report documented Facebook's role in genocide incitement | Demonstrates catastrophic failure of English-centric moderation |
| Ethiopia (2020–2022) | Delayed response to Amharic/Tigrinya content | Hate speech preceded violence; slow platform response | Low-resource language gaps enable real-world harm |
| India (2020–present) | Hindi/regional language misinformation | WhatsApp-mediated mob violence linked to unmoderated content | Encrypted + low-resource = compounded risk |
| Philippines (2022) | Tagalog election misinformation | Documented interference with limited platform response | Electoral integrity requires multilingual safety |
| Brazil (2022–2023) | Portuguese misinformation during elections | Platform gaps in Portuguese moderation capacity | Even "high-resource" languages face coverage gaps |

*Table 11.* Regulatory Approaches to AI Content Moderation by Region

| Region/Framework | Key Requirements | Multilingual Provisions | Gap Analysis |
|---|---|---|---|
| EU AI Act (2024) | Risk-based classification; transparency requirements | Implicit: "all Union languages" for high-risk systems | No explicit language-disaggregated reporting |
| EU DSA (2022) | Systemic risk assessment; researcher access | Requires assessment "per Member State" | Language proxied through geography |
| UK Online Safety Act | Illegal content duties; risk assessments | Wales/Scotland provisions | Limited low-resource language guidance |
| India IT Rules | Traceability; automated tools mandate | Hindi + English focus | 22 scheduled languages underserved |
| Singapore POFMA | Corrections for falsehoods | English, Mandarin, Malay, Tamil | Only 4 official languages covered |

*Table 12.* Defense Mechanisms Against Multilingual Jailbreaking: Effectiveness and Limitations

| Defense | Mechanism | English Effectiveness | Multilingual Effectiveness | Key Limitation |
|---|---|---|---|---|
| MART (Chao et al., 2024) | Multi-round automatic red-teaming | 84.7% reduction | Unknown | English-only evaluation |
| RPO (Agarwal et al., 2024) | Minimax defensive optimization | Near 0% ASR | Unknown | Requires language-specific tuning |
| PAT (Jain et al., 2024) | Prompt adversarial tuning | Near 0% ASR | Unknown | Guard prefix may not transfer |
| Safety Distillation (Üstün et al., 2024) | Pre-training safety embedding | 85% | 78% | Requires multilingual safety data |
| Translation Guardrails | Translate → English filter → translate back | N/A | Variable | Adds latency; loses context |
| Perplexity Filtering | Reject high-perplexity inputs | Moderate | Poor | Low-resource = high perplexity |

*Table 13.* Documented Bias Patterns in AI Content Moderation

| Bias Type | Finding | Affected Groups | Source |
|---|---|---|---|
| Racial bias in toxicity | AAE tweets 1.5× more likely to be flagged as toxic | African American speakers | (Sap et al., 2019) |
| Reclaimed language | Terms reclaimed by marginalized groups flagged as abusive | LGBTQ+, racialized minorities | (Davidson et al., 2019) |
| Dialect bias | Lower quality responses to AAE, Indian English, Appalachian English | Dialect speakers | (Hoffman et al., 2024) |
| Algorithmic arbitrariness | 38% conflicting predictions with different random seeds | All users; disparate impact on minorities | (Raji et al., 2024) |
| Geographic bias | Overrepresentation of developed countries in training data | Global South users | (Sabbir et al., 2024) |
| Annotation bias | Systematic differences across hate speech corpora | Depends on corpus origin | (Pamungkas et al., 2020) |

*Table 14.* Key Multilingual Safety Benchmarks and Datasets

| Benchmark | Languages | Focus | Limitations |
|---|---|---|---|
| MULTITuDE (Macko et al., 2023) | 11 | Machine-generated text detection | Limited low-resource coverage |
| BLUFF (Lucas et al., 2026b) | 58 low-resource | False and synthetic content detection | Concentrated on dis/misinformation |
| M-RewardBench (Gureja et al., 2025) | 23 | Reward model evaluation | Excludes very low-resource languages |
| AfroBench (Ojo et al., 2025) | 15 African | General LLM capabilities | Safety not primary focus |
| SEACrowd (Lovenia et al., 2024) | 50+ SE Asian | Multimodal, multilingual | Safety subset limited |
| Aya Red-Teaming (Aakanksha et al., 2024b) | 8 | Global/local harm distinction | Small language sample |
| M5 (Kim et al., 2024) | 41 | Multilingual multimodal | Safety not primary focus |
| HateXplain (Mathew et al., 2021) | 1 (English) | Explainable hate speech | English only |

*Table 15.* Tokenization Costs by Script and Language Family

| Script/Language | Tokens/Sentence | Cost Ratio vs. English | Example Model | Implication |
|---|---|---|---|---|
| English (Latin) | 54.1 | 1.0× | GPT-4 | Baseline |
| Spanish (Latin) | 67.3 | 1.2× | GPT-4 | Moderate overhead |
| Chinese (Hanzi) | 89.2 | 1.6× | GPT-4 | Character-level tokenization |
| Arabic (Arabic) | 112.4 | 2.1× | GPT-4 | Script complexity |
| Hindi (Devanagari) | 156.8 | 2.9× | GPT-4 | Subword fragmentation |
| Thai (Thai) | 201.3 | 3.7× | GPT-4 | No whitespace tokenization |
| Tibetan (Tibetan) | 305.4 | 5.6× | GPT-4 | Extreme fragmentation |

Data from Ahia et al. (2023). Costs compound: longer inputs = higher latency, cost, and category consumption.

