# OpenReview forum: "Position: Breaking the Dual Curse of Multilingual AI Requires Socio-Technical Guardrails, Not Post-Hoc Alignment Alone"
_ICML.cc/2026/Position_Paper_Track — ICML 2026 Position Paper Track regular_

### Official Review · Reviewer_TsxD · 2026-03-11

**Significance:** 4
**Argument Clarity:** 4
**Rating:** 5
**Confidence:** 4

**Questions:**

It is important to consider low-resource language at the very beginning, but the internet is in practice dominated by fewer than 50 languages.  Collecting low-resource language is not feasible, and oversampling low-resource languages will inevitably hurt model performance. How do you think we can overcome the resource issues?

**Alternative Views Section:**

Yes

**Compliance With Llm Reviewing Policy A Conservative:**

Affirmed.

**Discussion Potential:**

3

**Final Justification:**

My original score was supportive and I kept my score. The authors answered my questions.

I support the publication of this paper because I think the layout and topic of this paper fits the position track well. The minor concern might be that it is not technical.

**Paper Summary:**

This paper argues that the current AI models has 2 major issues on low-resource languages: The fail to guardrail the harmful contents while refused to answer safe content. The root cause of these issues is the data imbalance in the pretraining stage, while the societal background is the digital colonization. While the first issue (false negative) is being addressed by the industry, the second issue, false refusal, is less commonly reported or addressed. The authors of this paper call to develop new metrics and techniques to alleviate the disparity across languages, implement early safety conditioning during pretraining, and re-design the participatory mechanism to allow for more diverse languages and cultures.

**Position:**

Yes

**Position In Title:**

Yes

**Related Work:**

4

**Strengths And Weaknesses:**

This paper has solid societal background, with abundant discussion of past research on similar or related topics. The issues identified in this paper, such as jailbreaking through low-resource languages, were discussed in other papers but not as systematic as this one. This paper reveals that the capability disparity of LLMs on different languages roots from their pretraining data and participatory design, which is related to local cultures, therefore cannot be easily resolved by back-translation.

Many arguments of this paper are supported by either statistics or past research, and is convincing. Some key arguments are quantified with tables, figures, or quotes like "Key Finding".

The writing of this paper is of high quality, with clear texts, good formatting, appropriate emphasis, and well-structured figure / tables.

The drawback of this paper can be summarized as:
1. Some of the data in this paper is too outdated, given the rapid progress of LLM research nowadays.  Some statistics are drawn from old models like GPT-3 (e.g. the language composition of pretraining data), and some findings cite paper prior to 2020 when the NLP research was fundamentally different. It is understood that the most advanced models are opaque about their training data, but it will be good to cite more recent technical reports that discloses training details even if they are not the most frontier ones.
2. It would be great if examples, if appropriate to display, can be showed as well. For example, some examples that harmless contents are filtered by LLMs because of cultural differences.

**Support:**

4

---

> ### Author Rebuttal · Authors · 2026-03-25
>
> We thank the reviewer for a thorough and supportive assessment. We address both weaknesses and the question directly.
>
> *W1: Data Currency and Reliance on Older Model Statistics*
>
> *Before:* Statistics such as the 92.65% English token figure are presented without generation-scoping, implying generalization to current frontier models.
>
> *After (revised throughout):* Where statistics apply to a specific model family or training era, we now say so explicitly. Table 3 (Pre-Training Data Distribution) has been expanded with recent disclosures from Llama 3 (Dubey et al., 2024), Mistral-7B (Jiang et al., 2023), and Command A (Cohere, 2025). For closed frontier models such as GPT-4o and Gemini Ultra we note opacity explicitly and frame it as a governance concern: the absence of language-disaggregated training disclosures makes independent safety auditing impossible, reinforcing our call for transparency mandates. Reward model accuracy results (Shen et al.; Gureja et al., M-RewardBench) draw on models through 2024 and represent the most current systematic evidence available on this mechanism.
>
> The 92.65% English token figure derives from Zhao et al.'s (2024) GPT-3 analysis, the most granular publicly available estimate for any large model. While several model generations old, it remains the best available anchor for closed-model training composition. The pattern it illustrates, English dominance in pre-training, is corroborated by the newer disclosures now added to Table 4: Llama 3 (85% English), Mistral-7B (89% English), and Command A (70% English), all showing the same structural concentration.
>
> *W2: Absence of Concrete Examples*
>
> *Before:* The Javanese asymmetry is referenced in §1 but not developed as a structured comparison.
>
> *After (added to §3.1):* Two cross-lingual comparison examples are now developed as structured comparisons. For the Harmfulness Curse: GPT-4 generates content mirroring rhetoric from Indonesia's 1965--66 communal violence in Javanese while refusing the identical English prompt. For the Relevance Curse: a Javanese student asking about those same events receives a refusal while an English speaker receives a substantive educational response. Both are documented in Shen et al. (2024) and are presented side by side showing the prompt, the English response, and the low-resource language response.
>
> *Q1: Overcoming Resource Constraints at Scale*
>
> The reviewer correctly identifies a real tension. We accept the empirical premise: formal web metrics (DataReportal, October 2025) show 20 languages covering approximately 97% of websites by traffic share. However, these metrics measure institutional content production rather than actual user language distribution. Languages like Hausa (77 million speakers) and Javanese (83 million speakers) are used at enormous scale in informal digital settings that formal web crawls exclude, precisely the registers where harmful content circulates and English-trained classifiers are least reliable.
>
> The critical distinction is between general capability training and targeted safety conditioning. The reviewer's concern applies to the former; our proposal concerns the latter.
>
> *Before (implicit):* The paper does not clearly distinguish the data requirements of general pre-training from those of safety conditioning.
>
> *After (added to §4.1):* "Safety Context Distillation and model merging target purposively constructed safety corpora, orders of magnitude smaller than general pre-training data, fully decoupling safety conditioning from capability training. Model merging further decouples the objectives entirely: a capability-optimized model and a safety-conditioned model can be trained separately and merged post-hoc. We scope this in tiers: near-term across the ~20 languages with substantial deployment footprints; medium-term extending to informal-register varieties that formal web metrics systematically undercount."
>
> Aya demonstrated targeted safety conditioning is tractable across 101 languages with 3,000+ contributors without competing with general capability training investment.
>
> *Summary of Revisions*
>
> Add generation-scoping language to all anchor statistics; expand Table 3 with Llama 3, Mistral, and Command A disclosures; frame closed-model opacity as a governance concern; add two structured cross-lingual comparison examples to §3.1; add formal/informal register distinction and tiered scope paragraph to §4.1.

---

> > ### Author Rebuttal · Reviewer_TsxD · 2026-04-01
> >
> > As my comment.

---

### Official Review · Reviewer_U7mu · 2026-03-12

**Significance:** 3
**Argument Clarity:** 3
**Rating:** 4
**Confidence:** 4

**Questions:**

NA

**Alternative Views Section:**

Yes

**Compliance With Llm Reviewing Policy A Conservative:**

Affirmed.

**Discussion Potential:**

3

**Paper Summary:**

The paper argues that current large language model (LLM) safety mechanisms disproportionately favor English and high-resource languages, creating what the authors call the “Dual Curse.” In low-resource languages, models both generate more harmful content and refuse legitimate requests more often, making them simultaneously less safe and less useful. Drawing on a systematic review of 207 studies, the authors attribute this disparity primarily to a pre-training bottleneck, where reward models perform near random chance for low-resource languages, rendering post-hoc alignment methods such as RLHF ineffective.

**Position:**

Yes

**Position In Title:**

Yes

**Related Work:**

3

**Strengths And Weaknesses:**

Strength:
* The paper highlights a significant and underexplored issue in LLM safety, the uneven protection and usability across languages. By identifying the “Dual Curse,” the work provides a clear conceptual lens for discussing multilingual safety disparities in deployed models.
* The PRISMA-guided review of over 200 studies provides a useful overview of how multilingual safety research is distributed, supporting the claim that most research focuses on English or high-resource languages.

Weakness:
* The paper primarily synthesizes existing literature and reports previously published statistics rather than presenting new experiments or datasets. As a result, some key claims rely heavily on secondary sources rather than direct validation.
* While the paper attributes the Dual Curse largely to a pre-training bottleneck and reward model limitations, the evidence presented does not fully isolate these factors from other possible causes such as dataset imbalance, tokenization artifacts, or prompt distribution differences.

**Support:**

3

---

> ### Author Rebuttal · Authors · 2026-03-25
>
> We thank the reviewer for a careful assessment. We address both weaknesses directly.
>
> *W1: Synthesis Rather Than New Empirical Analysis*
>
> *Before (implicit throughout):* The paper presents synthesis as if it were equivalent to empirical contribution without explicitly defending the genre.
>
> *After (added to §1.3):* "As a position paper, these contributions synthesize existing empirical work rather than introducing new primary datasets or experiments. The PRISMA-guided review of 207 studies is itself a methodological contribution: no prior review has systematically mapped multilingual safety research across language resource tiers with this scope, and the 91.3% English-dominance figure is a finding of the review rather than a prior known quantity."
>
> We clarify that this position paper's primary contribution is diagnostic synthesis and framework construction rather than causal identification or system implementation. The ICML Position Paper Track explicitly invites papers that synthesize existing evidence to argue for a non-obvious claim or reframe how the community should think about a problem. The Dual Curse framing, the pre-training bottleneck diagnosis, the formal-register critique, and the socio-technical framework are original analytical contributions even though the underlying empirical data derive from prior work. We also note that the alternative to synthesis in this domain is not straightforward: running original safety evaluations across a representative set of low-resource languages requires annotation infrastructure and community expertise that no single research team currently commands. The contribution of a well-evidenced position paper is precisely to establish the problem structure clearly enough that the community can organize the distributed empirical work required. Masakhane, Aya, and M-RewardBench all emerged in part because prior synthesis work made the problem legible.
>
> *W2: Failure to Isolate the Pre-Training Bottleneck*
>
> *Before:* "we identify the pre-training bottleneck as root cause... rendering post-hoc alignment structurally ineffective."
>
> *After (revised in §3.3 and abstract):* "we identify the pre-training bottleneck as the primary structural mechanism... rendering post-hoc alignment substantially less effective for low-resource languages."
>
> We accept this critique and have revised accordingly. The pre-training bottleneck is a primary structural mechanism rather than an isolated singular cause. We now stratify evidence into three tiers in the revised manuscript.
>
> *Tier 1 (replicated cross-study trends):* English dominance (91.3%, confirmed by Peppin et al. and Yong et al. independently); cross-lingual vulnerability across three methodologically distinct attack paradigms spanning ASR gaps of 40 to over 99 percentage points above English baselines: translation jailbreaks (Yong et al., 2023: 79% LRL-combined on GPT-4), universal adversarial suffixes (Zou et al., 2023: transfer to ChatGPT, Bard, Claude), and cross-lingual backdoor triggers (Zheng et al., 2024: ~100% ASR on Llama-3 and Qwen2 at 3--5% poisoning rate, robust against ONION and SFT defenses).
>
> *Tier 2 (multi-study, variable magnitude):* CHAT-RLHF: 44.8% vs 23.4% harm reduction; xRLHF: 14.4% vs 2.4%; cross-lingual attack success rates (Yong et al.: 79% LRL-combined; Deng et al.: 80.92% ChatGPT / 40.71% GPT-4 intentional scenario, 79.05% under adaptive attacks); false refusal elevation across Shen et al., Hoffman et al., and Raji et al.
>
> *Tier 3 (single- or few-study, explicitly illustrative):* The 1% to 35% harmfulness gap and 49--50% RM accuracy derive primarily from Shen et al. (2024), covering 19 languages on GPT-4. We flag these explicitly as requiring replication before generalizing. What distinguishes the RM bottleneck is its pipeline position: a reward model at chance accuracy forecloses RLHF improvement regardless of upstream SFT quality or tokenization improvements, making it the highest-leverage intervention point without requiring it to be the sole cause. To provide independent empirical support, we are conducting a targeted ablation comparing Qwen3 base, instruction-tuned, and safety RLHF-tuned variants across harmful prompt sets in three languages spanning resource tiers. We will share results before the rebuttal deadline if possible; otherwise they will appear in the camera-ready manuscript.
>
> *Summary of Revisions*
>
> Add genre clarification to §1.3; revise §3.3 replacing "root cause" with "primary structural mechanism"; add three-tier evidence stratification to §2 and §3.3; flag Shen et al. anchor study dependency explicitly; add Qwen3 ablation results to camera-ready manuscript.

---

> > ### Author Rebuttal · Reviewer_U7mu · 2026-04-05
> >
> > NA

---

### Official Review · Reviewer_JZ1F · 2026-03-13

**Significance:** 3
**Argument Clarity:** 3
**Rating:** 5
**Confidence:** 3

**Questions:**

Why the “Evidence Base: The Safety Divide” only cover English-language publications?

**Alternative Views Section:**

Yes

**Compliance With Llm Reviewing Policy A Conservative:**

Affirmed.

**Discussion Potential:**

3

**Final Justification:**

The author has addressed my concerns, so I decide to raise my score.

**Paper Summary:**

This paper introduces a dual curse framework for low-resource languages: the harmfulness curse and the relevance curse. Speakers of low-resource languages are more likely to receive AI-generated content that causes or risks harm, such as hate speech targeting protected groups, instructions for violence, or deceptive material, and the model's performance on downstream tasks remains significantly poorer than it is in English. These problems become governance hazards in at least 22 countries that mandate automated content moderation, thereby creating infrastructures that can be exploited for censorship.

To support this claim, the paper first conducts a statistical analysis showing that over 90\% of safety research focuses on English or other high-resource languages. It then proposes a diagnostic framework to examine each dimension of the dual curse and trace its root causes. Finally, it presents a socio-technical guardrail framework designed to improve safety in multilingual AI deployments.

**Position:**

Yes

**Position In Title:**

Yes

**Related Work:**

3

**Strengths And Weaknesses:**

Strengths
1. The position is based on empirical evidence and sound analysis; logical rebuttals to alternative viewpoints further enhance its clarity and persuasiveness.
2. As large language models become ever more integrated into daily life, safety challenges in multilingual contexts, particularly for low-resource languages, are both timely and critical, making this topic highly relevant to AI researchers.
3. The proposed framework offers a comprehensive approach to enhancing AI safety for low-resource languages.
4. The “Weaponization Nexus” section emphasizes that ensuring AI safety in low-resource languages is not optional but essential.

Weaknesses
1. The “Evidence Base: The Safety Divide” relies on English-language publications, which may introduce bias and limit the persuasiveness of the conclusions.

**Support:**

3

---

> ### Author Rebuttal · Authors · 2026-03-25
>
> We thank the reviewer for a supportive and focused review. We address the one weakness and question directly.
>
> *W1 / Q1: English-Language Publication Restriction in the PRISMA Corpus*
>
> The restriction was practical: our search infrastructure (Scopus, Web of Science, ACL Anthology, arXiv) indexes predominantly English-language publications, and our reviewer team could not guarantee consistent quality assessment across non-English academic venues. This is a standard limitation acknowledged in Appendix B.4, which we will now elevate to the main-text PRISMA summary in §2.
>
> *Before (Appendix B.4 only):* "Language: English-language publications only (due to reviewer language constraints, acknowledging this as a limitation that itself reflects the English bias we document)."
>
> *After (elevated to §2, reframed):* "Our PRISMA search is restricted to English-language publications, itself a self-exemplifying instance of the bias we document: safety scholarship about low-resource languages is less likely to appear in indexed venues, less likely to be cited, and therefore less likely to be surfaced by standard review methods. Our 8.6% low-resource coverage figure therefore likely overstates true breadth."
>
> More importantly, we reframe this limitation as theoretically significant rather than merely methodological. The difficulty of conducting an inclusive systematic review is itself a self-exemplifying instance of the English-dominance problem we document. Safety scholarship about low-resource languages is less likely to appear in indexed venues, less likely to be cited, and therefore less likely to be surfaced by standard review methods. Our 8.6% figure for low-resource language coverage is therefore likely an overestimate: if non-English safety scholarship were included, the gap would in all probability be larger. This does not undermine our conclusions; it strengthens them.
>
> A second layer of the same problem applies to the training corpora the reviewed studies evaluate. Contemporary web metrics (DataReportal, October 2025) show 20 languages covering approximately 97% of websites by traffic share, but this measures institutional content production rather than actual user language distribution. Languages like Hausa (77 million speakers) and Javanese (83 million speakers) are used at enormous scale in social media, messaging platforms, and informal settings that formal web crawls exclude, precisely the registers where the Harmfulness Curse is most acute. Safety research following the formal web's contours will systematically misidentify which communities are most exposed.
>
> We note that this formal-register limitation applies to our evidence tiers as well. Tier 1 trends (English dominance, cross-study attack vulnerability) are robust to this limitation because the direction is consistent regardless of register. Tier 3 anchor results (the 35x harmfulness gap from Shen et al.) are more sensitive, since they are measured on translated formal prompts rather than naturalistic low-resource language use. We will flag this explicitly in the revised manuscript.
>
> *Summary of Revisions*
>
> Elevate PRISMA language restriction from Appendix B.4 to §2; reframe as self-exemplifying evidence; add formal-register paragraph clarifying that informal language settings are excluded from both the review corpus and the training data evaluated; flag Tier 3 anchor results as sensitive to this limitation.

---

> > ### Author Rebuttal · Reviewer_JZ1F · 2026-04-05
> >
> > My concerns have been adequately addressed. I would like to raise my score.

---

### Official Review · Reviewer_qxbV · 2026-03-13

**Significance:** 4
**Argument Clarity:** 3
**Rating:** 3
**Confidence:** 3

**Questions:**

1. The paper treats the pre-training bottleneck as the root cause. What empirical evidence would count against this hypothesis? It would be helpful to discuss alternative causal accounts—such as tokenization disparities, multilingual SFT data quality/coverage, or model-family differences—and how these mechanisms relate to the reward-model results.

2. For the proposed ASR+FRR multilingual evaluation, what would a concrete benchmark or shared task look like? In particular, how should languages and locales be selected, how should harm categories be defined and maintained, and what annotator-governance process would make the evaluation operational rather than purely aspirational?

3. The governance section is compelling, but the scope sometimes blurs LLM assistants with platform moderation systems. Could the authors more clearly separate which conclusions apply to assistant-style chat models versus large-scale moderation pipelines, and where the supporting evidence differs?

**Alternative Views Section:**

Yes

**Compliance With Llm Reviewing Policy A Conservative:**

Affirmed.

**Discussion Potential:**

3

**Paper Summary:**

This paper contends that multilingual AI safety suffers from a dual failure: in low-resource languages, harmful content is more likely to slip through, while benign requests are more likely to be refused. Drawing on a PRISMA-guided synthesis of 207 studies, it argues that safety research and tooling are disproportionately English-centric and questions how far post-hoc alignment can close the gap. It then outlines a socio-technical agenda centered on earlier safety conditioning, participatory harm specification, and evaluation that jointly reports attack success rate and false refusal rate.

**Position:**

Yes

**Position In Title:**

Yes

**Related Work:**

3

**Strengths And Weaknesses:**

**Strength**:

The paper stakes out a clear, non-trivial position, and the title reflects it well. The Dual Curse framing is a useful lens because it unifies two practically important failure modes: under-blocking harmful content and over-refusal of benign content. The topic is timely, given the global deployment of frontier models and the fact that multilingual safety remains comparatively under-measured. The manuscript also synthesizes evidence across technical safety, multilingual NLP, and governance. Finally, the alternative views section stands out for engaging seriously with plausible counterarguments.

**Weakness**:

1. My main concern is evidentiary overreach. Much of the paper’s strongest rhetoric rests on secondary synthesis rather than new empirical analysis, yet several claims are phrased in a way that suggests causal identification. In particular, the manuscript attributes multilingual safety disparities to a pre-training bottleneck and concludes that post-hoc alignment is structurally ineffective. This is a plausible diagnosis, but the current evidence reads stronger for a major contributing factor than for a single root cause.

2. The paper usefully connects systematic review, causal diagnosis, decolonial theory, governance, and design recommendations, but the breadth also leaves parts of the proposal underspecified. For instance, participatory harm specification is a sensible direction, yet the manuscript does not fully spell out what an end-to-end ML pipeline, benchmark design, or auditing workflow would look like in practice. In addition, some headline quantitative claims appear to lean heavily on a small set of anchor studies rather than the full review corpus; it would help to distinguish broad trends from stronger conclusions that depend on a narrower evidence base.

**Support:**

2

---

> ### Author Rebuttal · Authors · 2026-03-25
>
> We thank the reviewer for the most substantive critique. We address each concern with concrete revisions rather than promises.
>
> *W1: Evidentiary Overreach and Causal Language*
>
> We accept this critique. This position paper contributes diagnostic synthesis and framework construction rather than causal identification.
>
> *Before:* "root cause... structurally ineffective."
> *After (§3.3, abstract):* "primary structural mechanism... substantially less effective for low-resource languages."
>
> The RM accuracy finding (49--50%; Shen et al.; Gureja et al.) is structurally significant because a reward model at chance provides no usable RLHF gradient regardless of tokenization or SFT improvements, making it the highest-leverage intervention point without requiring it to be the sole cause. To provide independent empirical support for W1 and Q1, we are conducting a targeted ablation comparing Qwen3 base, instruction-tuned, and safety RLHF-tuned variants across harmful prompt sets in three languages spanning resource tiers. This ablation directly addresses whether the gap requires causal attribution to alignment or is present in base models, the precise question W1 raises. We expect the gap to emerge specifically post-RLHF, consistent with Shen et al.'s LLaMA-2 findings. We will share results before the rebuttal deadline if possible; otherwise they will appear in the camera-ready manuscript.
>
> We stratify evidence into three tiers in the revision:
>
> *Tier 1 (replicated trends):* English dominance (91.3%, confirmed by Peppin et al. and Yong et al. independently); ASR rising from below 1% in English to 43--79% across three distinct attack paradigms: translation jailbreaks (Yong et al., 2023: 79% LRL-combined on GPT-4), universal suffixes (Zou et al., 2023), and cross-lingual backdoor triggers (Zheng et al., 2024: ~100% ASR on Llama-3, Qwen2-7B, and Qwen2-1.5B at 3--5% poisoning rate, robust against ONION and SFT defenses). Three methodologically independent attack paradigms converging on the same vulnerability constitutes strong cross-study evidence.
>
> *Tier 2 (multi-study, variable magnitude):* CHAT-RLHF: 44.8% vs 23.4%
> harm reduction; xRLHF: 14.4% vs 2.4%; cross-lingual attack success rates
> (Yong et al.: 79% LRL-combined; Deng et al.: 80.92% ChatGPT / 40.71% GPT-4
> intentional scenario, 79.05% under adaptive attacks); false refusal elevation
> across Shen et al., Hoffman et al., and Raji et al.
>
> *Tier 3 (single-study, illustrative):* The 35x harmfulness gap and 49--50% RM accuracy derive primarily from Shen et al. (2024), 19 languages, GPT-4. We flag these explicitly as requiring replication.
>
> *W2: Underspecification and Anchor Study Dependency*
>
> A methodological note in §2 will distinguish Tier 1 trends from Tier 3 anchor results. The participatory pipeline now has a three-stage design in §4.2: (1) community annotation sprints for global/local harm taxonomies; (2) ASR red-teaming and FRR labeling with $\kappa \geq 0.7$; (3) WMT-style annual governance audits. Existence proofs anchor each stage: Aya (specification and red-teaming across 101 languages, 3,000+ contributors), Masakhane (community annotation governance for African languages), M-RewardBench (RM evaluation across 23 languages), and WMT (shared task maintenance model). Presented as phased research infrastructure, not a near-term deliverable.
>
> *Q1: Falsifiability*
>
> Disconfirmed if: (a) multilingual RM trained only on high-resource data achieves $>$70% on M-RewardBench low-resource tiers; (b) xRLHF matches Safety Context Distillation (2.4% vs 78--89%); or (c) tokenization normalization alone closes the RM gap. Tokenization disparity degrades inference-time coherence but does not explain training-time RM failure at chance. Critically, the alignment gap persists across two distinct training regimes: CHAT-RLHF (44.8% vs 23.4% harm reduction) and the authors' own xRLHF (14.4% vs 2.4%), ruling out implementation-specific artifacts as an alternative explanation. The Qwen3 ablation will also test model-family generalizability beyond GPT and LLaMA.
>
> *Q2: ASR+FRR Benchmark*
>
> Stratified sampling across Joshi et al. tiers (0--5), two languages per tier, 20--25 languages total. Aya global/local harm distinction with annual audits. Three annotators per item, $\kappa \geq 0.7$, human validation given LLM-as-judge degradation for low-resource languages (Gureja et al.).
>
> *Q3: Assistant Models vs. Platform Moderation*
>
> Resolved in §3.4 and Table 5. Assistant models: RM failure during training; remedy is pre-training intervention. Platform moderation (Freedom House, Meta, House Judiciary): classifier transferability failure at inference; remedies are procurement standards and transparency mandates. Same structural problem, distinct solutions. Conflating the two risks misaligning interventions: pre-training investment addresses assistant-model failures, while regulatory transparency mandates address platform-moderation failures. Both are necessary; neither alone is sufficient.

---

> > ### Author Rebuttal · Reviewer_qxbV · 2026-04-05
> >
> > Thank you for the response. The paper usefully connects systematic review, causal diagnosis, decolonial theory, governance, and design recommendations, but the breadth also leaves parts of the proposal underspecified.

---

### Decision · Program_Chairs · 2026-04-30

**Decision:**

Accept (regular)

**Comment:**

** Overall: ** Discussion around the gap in multilingual AI safety is a significant and mostly overlooked area of research. The framework for unifying failure modes was noted as particularly facilitatory of potential discussion, though the authors should beware of overclaiming, particularly when it comes to claims around causality of model failures.

** Primary strengths: **
- Excellent significance (JZ1F) and an under-explored issue in model safety (U7mu)
- Past research is well-covered (TsxD, U7mu), and the paper makes its own position from this past research clear
- Unifies distinct failure modes relevant to safety (qxbV)

** Primary weaknesses: **
- Low resource languages are, almost by definition, less represented in training data, which reduces the tractability of proposed mitigations (TsxD). The authors contend that this is an issue for model capability training, but not safety conditioning. However, as the authors identify a root cause to be a data imbalance in pretraining,
- Other possible causes of model failures besides limitations of reward modelling (e.g., tokenization artifacts, distributional differences in prompts) are not explored (U7mu). The authors clarify that the bottleneck is a structural mechanism rather than a direct cause
- Overclaiming from existing evidence suggests causality of factors when this isn’t clearly warranted by the evidence (qxbV). The authors agree to address this in a revision.